# Cryo-EM structure of the CDK2-cyclin A-CDC25A complex

Rhianna J. Rowland [1], Svitlana Korolchuk[1,4], Marco Salamina [1,5], Natalie J. Tatum [1], James R. Ault[2], Sam Hart [3], Johan P. Turkenburg [3], James N. Blaza [3], Martin E. M. Noble [1] ✉ & Jane A. Endicott [1] ✉

The cell division cycle 25 phosphatases CDC25A, B and C regulate cell cycle transitions by dephosphorylating residues in the conserved glycine-rich loop of CDKs to activate their activity. Here, we present the cryo-EM structure of CDK2-cyclin A in complex with CDC25A at 2.7 Å resolution, providing a detailed structural analysis of the overall complex architecture and key protein-protein interactions that underpin this 86 kDa complex. We further identify a CDC25A C-terminal helix that is critical for complex formation. Sequence conservation analysis suggests CDK1/2-cyclin A, CDK1-cyclin B and CDK2/3-cyclin E are suitable binding partners for CDC25A, whilst CDK4/6-cyclin D complexes appear unlikely substrates. A comparative structural analysis of CDK-containing complexes also confirms the functional importance of the conserved CDK1/2 GDSEID motif. This structure improves our understanding of the roles of CDC25 phosphatases in CDK regulation and may inform the development of CDC25-targeting anticancer strategies.

The cyclin-dependent protein kinases (CDKs) CDK1, CDK2 and CDK4/6 collectively regulate the eukaryotic cell cycle. They require association with their cognate cyclin partner for activity and are additionally controlled both by protein binding and post-translational modifications (reviewed in refs. 1–3). Phosphorylation of a conserved tyrosine (equivalent to Tyr15 in human CDK1) in all eukaryotes, and additional phosphorylation of an adjacent threonine (Thr14) in metazoans, located within the glycine-rich loop (G-loop) of the ATP binding site, inhibits CDK1/2 activity. CDC25 (M-phase inducer phosphatase) was identified in *Schizosaccharomyces pombe* and shown to promote entry into mitosis by dephosphorylating CDK1 Tyr15 within the G-loop[4,5]. In human cells, there are three CDC25 isoforms (named A, B and C[6–10]) that antagonise the activities of both Wee1 kinase that phosphorylates CDK1[11] and CDK2 on Tyr15[12], and Myt1 kinase that phosphorylates CDK1 on Thr14 and Tyr15[13–15]. Though early studies suggested that the CDC25 isoforms may have discrete and non-overlapping roles, the current model proposes they cooperate to regulate the major cell cycle transitions (reviewed in refs. 16–18). As cells transition into mitosis, CDC25B first activates CDK1-cyclin B at centrosomes in prophase[19–22], and then all three isoforms cooperate to amplify CDK1 activity in the nucleus[23–25]. At G1/S CDC25A activity is most significant in dephosphorylating and activating both CDK2-cyclin A and -cyclin E complexes[26–28].

CDC25 is an essential component in the response to damaged or unreplicated DNA. This role was first recognised in fission yeast, where it was demonstrated to be phosphorylated by CHK1 in response to DNA damage[29]. This pathway was subsequently found to be fundamentally conserved in mammalian cells[30,31]. A substantial body of work has since elaborated the pathways that impact CDC25 activity and expression to regulate the cell's response to damaged and unreplicated DNA at different stages of the cell cycle. At the molecular level, these checkpoint pathways regulate CDC25 activity through multiple mechanisms that include phosphorylation, proteolysis, binding to

[1]Translational and Clinical Research Institute, Newcastle University Centre for Cancer, Newcastle University, Paul O'Gorman Building, Framlington Place, Newcastle upon Tyne NE2 4HH, UK. [2]Astbury Centre for Structural Molecular Biology, School of Molecular and Cellular Biology, University of Leeds, Leeds LS2 9JT, UK. [3]York Structural Biology Laboratory and York Biomedical Research Institute, Department of Chemistry, University of York, Heslington, York YO10 5DD, UK. [4]Present address: Fujifilm, Belasis Ave, Stockton-on-Tees, Billingham TS23 1LH, UK. [5]Present address: Evotec (UK) Ltd., Milton, Abingdon OX14 4RZ, UK. ✉e-mail: martin.noble@ncl.ac.uk; jane.endicott@ncl.ac.uk

14-3-3 proteins and changes in sub-cellular localisation (reviewed in refs. 32–36).

The majority of the post-translational modification sites that regulate CDC25 activity are located within the unstructured, extended N-terminal sequence that diverges between CDC25 isoforms and precedes their conserved C-terminal catalytic domain[37,38]. CDC25s are not only CDK regulators but also CDK substrates and can be recruited to cyclins A, B, D and E via an RXL motif (where X is any amino acid, CDC25A residues 11-15 RRLLF[39]) that is conserved in other CDK substrates and inhibitors (reviewed in ref. 3). The association of CDC25 with 14-3-3 dimers is also mediated by residues located in the N-terminal sequence of CDC25 (reviewed in ref. 36). 14-3-3 binding has been proposed to directly inhibit CDC25 activity by masking the CDC25 cyclin binding motifs[40] and indirectly by modulating its cellular location (reviewed in ref. 41). Phosphorylation site mutants have identified two CHK1 phosphorylation sites on CDC25A, Ser178 (N-terminal region) and Thr507 (within the C-terminal tail), that mediate 14-3-3 binding[40,42]. The CDC25 C-terminal tail is also an important element of CDK-cyclin recognition and sequence differences between CDC25 isoforms have been shown to modulate CDC25 catalytic activity[43].

Early studies on CDC25A and B defined them as oncogenes[44], with changes to CDC25 expression observed in a variety of cancers (reviewed in refs. 36,45,46). Given their roles in regulating the cell cycle and response to checkpoint pathway signalling, small molecule inhibitors that target the active site have been reported (reviewed in ref. 47), and alternative allosteric approaches targeting CDC25-protein interactions have been considered[48]. However, there remains a lack of structural information on the binding of CDC25 proteins with their CDK-cyclin substrates, which likely hinders the development of CDC25 inhibitors and CDC25-targetting anticancer strategies. Whilst the catalytic domain structures of CDC25A (PDB 1C25[37]), CDC25B (PDB 1CWR[38]) and CDC25C (PDB 3OP3) have been solved, a structure of any CDC25 isoform in complex with a CDK-cyclin module is yet to be elucidated.

Here, we present the cryogenic electron microscopy (cryo-EM) structure of CDK2-cyclin A in complex with CDC25A at 2.7 Å global resolution. Together these proteins constitute an ~86 kDa complex, making it among the smallest asymmetric particles to have been determined by cryo-EM to a resolution better than 3 Å. This structure reveals the overall binding mode of CDC25A with CDK2-cyclin A, detailing interactions between the CDC25A catalytic domain and the N- and C-terminal lobes of CDK2. We identify a CDC25A C-terminal helix that binds at the CDK2-cyclin A interface and is critical for trimeric complex formation. Sequence conservation analysis against the cell-cycle CDKs and cyclins (CDK2 vs 1, 3, 4 and 6 with cyclin A vs B, E, D1 and D3) suggests CDK1/2-cyclin A, CDK1-cyclin B and CDK2/3-cyclin E are suitable binding partners for CDC25A, whereas CDK4/6-cyclin D sequences diverge at key interaction sites and appear unlikely substrates for CDC25A. These results support a model in which CDK4 and CDK6 are not regulated by G-loop phosphorylation or are dephosphorylated by other means. Comparisons with existing structures of Thr160-phosphorylated CDK2 (pT160CDK2) bound to kinase-associated phosphatase ((KAP) PDB 1FQ1[49]) and to cyclin-dependent protein kinase subunit 1 ((CKS1) PDB 1BUH[50]) indicate the GDSEID motif (single letter amino acid code, residues 205-210 in CDK2), which is conserved in CDKs 1-3 and is located in the CDK C-terminal lobe, is a mutual binding site for these proteins that also imparts selectivity, highlighting its importance as a CDK regulatory site.

## Results and discussion

### Data collection and structure determination

The CDK2-cyclin A-CDC25A complex was phosphorylated on CDK2 Tyr15 and Thr160, and the CDC25A was catalytically inactivated by mutating Cys431 to serine to generate a stable ternary complex. (See Supplementary Methods and Supplementary Fig. 1 for further details). Preliminary data collection on a 200 kV Glacios readily yielded promising 2D class averages, (Fig. 1a). However, significant preferential orientation was evident (Fourier space coverage ~10% estimated by Cryo-EF[51]) that limited 3D refinement (Supplementary Fig. 2a–c). Particle picking with Topaz[52,53] improved the number of unique 2D views, (Fig. 1a, b), yielding a reconstruction that encompassed the full complex. However, the map exhibited considerable directional resolution anisotropy, visualised by streaking in the EM density (Supplementary Fig. 2d–f).

To improve the particle orientation distribution, the complex was spiked with CHAPS at concentrations ranging 0.1 – 1.0 X CMC (critical micelle concentration) immediately prior to grid preparation. The grids were screened on a 200 kV Glacios and the CHAPS concentration most beneficial for improving the particle orientation was evaluated from the 2D class averages. The beneficial effect of CHAPS was evident at all concentrations tested; however, a wider variety of unique 2D views were identified using 0.5 and 1.0 X CMC CHAPS (Fig. 1c, d).

High-resolution data collection on grids prepared with 0.5 and 1.0 X CMC CHAPS was performed on a 300 kV FEI Titan Krios equipped with a BioQuantum K3 detector and energy filter (Supplementary Fig. 1c and Table 1). The full data processing workflow is described in Supplementary Fig. 3, but in summary, particles of the CDK2-cyclin A-CDC25A complex were picked using blob picker, 2D classified and sorted into 3D classes. Particles from the best 3D class were chosen to train Topaz for further picking[52]. Topaz-picked particles were 2D and 3D classified, with the best 3D class refined by homogeneous and non-uniform refinement. Per particle reference-based motion correction was performed and non-uniform refinement was repeated with the motion corrected particles to yield the final 3D reconstruction at 2.7 Å resolution constituting 670,852 particles. The resulting EM map (EMD-19408) showed clear density for the overall quaternary architecture, secondary structure, and side chain organisation of the trimeric complex (Fig. 1e and Supplementary Fig. 4).

ModelAngelo[54] was used for automated model building of the trimeric complex, completed by manual re-building of low-confidence regions in Coot[55] according to the EM map, followed by real-space refinement in Phenix[56,57] to generate the full CDK2-cyclin A-CDC25A model (PDB 8ROZ). The local resolution within the complex varies from 2.4 Å to 3.0 Å (Fig. 1f), with the highest resolution within the CDK2-cyclin A core, and the lowest resolution within the N-terminal β-sheet domain of CDK2 (residues 9-18) and flexible loop regions of CDC25A (residues 420-424, and 492-495). Nevertheless, all expressed residues for each protein (human CDK2 residues 1-298, bovine cyclin A2 residues 169-430 Uniprot P30274 (equivalent to human cyclin A residues 171-432, Uniprot P20248), and human CDC25A residues 335-524 (Fig. 2a) could be modelled.

### Overview of the CDK2-cyclin A-CDC25A complex

In the trimeric complex, CDK2 has its characteristic bi-lobal structure, comprising the β-sheet N-terminal and predominantly helical C-terminal lobes, both of which are generally well conserved with that of CDK2 in the CDK2-cyclin A complex (PDB 1JST[58], Fig. 2a, b). The structure of cyclin A is extremely well conserved, consisting of two helical domains (the N- and C-terminal cyclin box folds, N-CBF and C-CBF) essentially identical in chain topology that bind both lobes of CDK2 to form an extensive protein-protein interface (Fig. 2a, b). The CDC25A catalytic domain structure is broadly consistent with the monomeric crystal form (PDB 1C25[37]) comprising an α/β-domain with a central 5-stranded parallel β-sheet (of strand order 15423) enclosed by 5 α-helices (RMSD of aligned residues (335–524) = 0.9 Å), (Fig. 2c). The mutated catalytic residue (Cys431Ser) initiates the conserved PTP $C(X)_5R$ loop between a central β-strand (β4) and α-helix (α4), creating a shallow active site that recognises CDK2 pTyr15. The CDC25A catalytic domain bridges the bi-lobal structure of CDK2,

binding on the opposite face to cyclin A, to form an extensive but discontinuous interface of ~1260 Å$^2$ (Fig. 2a). In contrast to the CDC25A crystal structure which visibly terminates at Glu495, the CDC25A sequence can be visualised to Leu524 (Fig. 2c). This longer C-terminal tail coverage reveals a helix (α6), spanning residues 495–524, which binds at the CDK2-cyclin A interface relatively distant from the CDC25 core (Fig. 2a). This helix interacts with both CDK2 and cyclin A, providing a mutual binding site in the trimeric complex. Specifically, the CDC25A C-terminal helix interacts with the N-terminal αC helix (PSTAIRE, residues 45–51) and C-terminal activation segment (residues 145–172 between the DFG and APE motifs and includes phosphorylated Thr160) of CDK2, whilst providing the only point of contact for cyclin A. The interaction with the C-terminal cyclin box fold (C-CBF) of cyclin A is relatively remote, forming a small protein-protein interface of ~170 Å$^2$ mediated by the loop linking α3′ to α4′ in the C-CBF. The lack of electron density for these C-terminal residues in the existing CDC25A crystal structure, indicates this region of CDC25A may be disordered, transitioning to nascent secondary structure on binding the CDK2-cyclin A substrate.

Prior to our determination of the structure of the CDK2-cyclin A-CDC25A complex, we analysed it by hydrogen-deuterium exchange mass spectrometry (HDX-MS), comparing monomeric CDC25A against the trimeric complex (Supplementary Fig. 5 and Supplementary Table 2, PXD050866). In agreement with the structure, this analysis highlighted a region of CDC25A that is significantly protected on complex formation (Supplementary Fig. 5a–c); specifically, peptide 1 (VRERDRLGNEYPKLHYPEL, residues 445–463) which starts towards the middle of helix α4 C-terminal to the mutated catalytic Cys431Ser residue and continues into the succeeding loop. The VRERDRLGNEY and PKLHYPEL portions of peptide 1 show ~10% and ~15% difference in fractional uptake respectively. Although the latter portion of this

peptide sequence (residues 457-464, PKLYHPEL) is partially protected in the monomeric form by the most N-terminal CDC25A residues, the enhanced protection for the full peptide sequence (VRERDRLG-NEYPKLHYPEL) in the trimeric complex can be rationalised by its interaction with the CDK2 GDSEID motif (described in detail below), which binds above the C-terminal end of this central CDC25A helix. Supporting this CDK2-CDC25A interaction site, reciprocal HDX-MS analysis of CDK2 (CDK2-cyclin A vs the trimeric complex) (Supplementary Fig. 5d–f) revealed significant protection for CDK2 residues 196–220 (MVTRRALFPGDSEIDQLFRIFRTLG) within the C-terminal lobe, which include the GDSEID sequence that makes important interactions with CDC25A (detailed below).

Within the CDK2 N-terminal lobe, a peptide spanning residues 38–51 (DTETEGVPSTAIRE) shows significant protection in the ternary complex. This peptide includes the PSTAIRE (αC) helix and preceding loop, which bind the CDC25A C-terminal helix (α6) (described in detail below). Slight, but statistically significant protection is also observed for a peptide within the CDK2 activation segment (residues 163–167, VVTLW). This peptide is not directly involved in CDC25A contact, but the immediately preceding residues (and activation segment as a whole) bind the CDC25A C-terminal helix (discussed below).

Notably, the HDX-MS data did not reveal any significant protection for the C-terminal domain of CDC25A, which is surprising given the dominant feature of the CDC25A C-terminal (α6) helix in the cryo-EM map. Given the relatively distant nature of the interaction with the CDK2-cyclin A interface (the CDC25A C-terminal helix backbone sits ~11 Å away from the backbone structure of CDK2), it is hypothesised that the C-terminal helix is not substantially solvent-protected in the bound state. Additionally, high levels of exchange of C-terminal peptides were observed across all timepoints in both the unbound and bound states, which may be attributed to the structurally dynamic

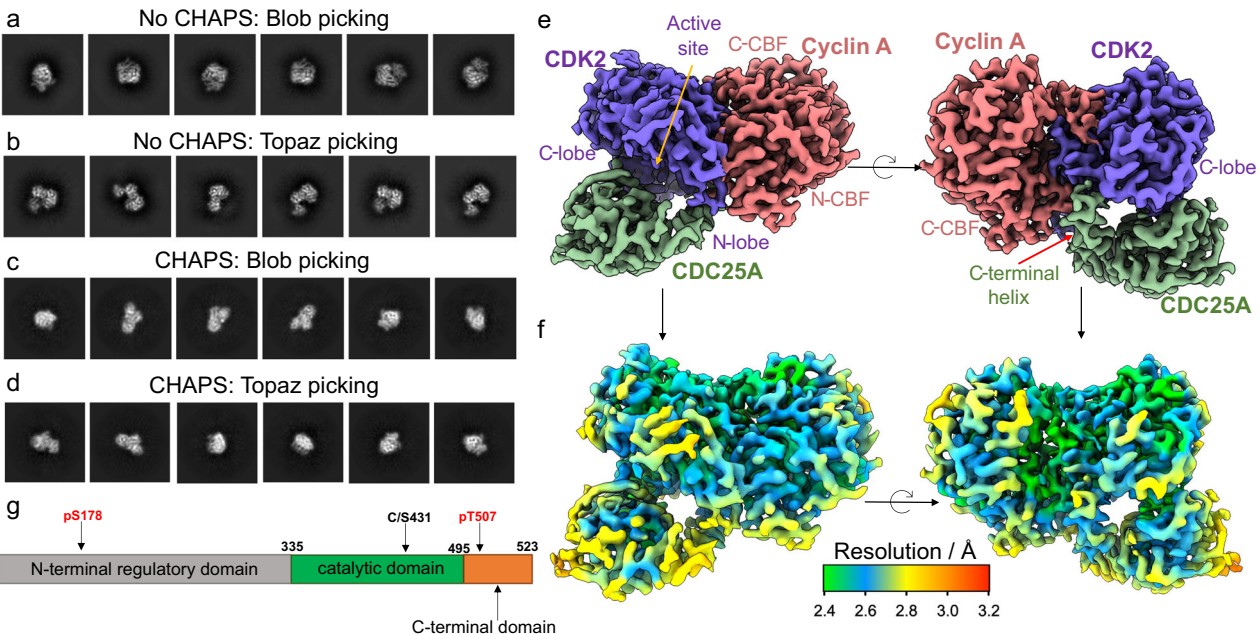

**Fig. 1 | Cryo-EM structure of the CDK2-cyclin A-CDC25A complex.**
**a–c** Representative 2D class averages of particles of the CDK2-cyclin A-CDC25A complex picked from preliminary data using (**a**) blob picking, showing significant preferential orientation in which most 2D classes represented a front-on view of CDK2-CDC25A and (**b**) Topaz picking, which improved the number of unique 2D views, but preferential orientation persisted. **c** During high-resolution data collection, the use of CHAPS significantly enhanced the particle orientation distribution, as indicated by the wider variety of 2D class averages observed from (**c**) blob picked and (**d**) Topaz picked particles. **e, f** Cryo-EM map of the CDK2-cyclin A-CDC25A complex at 2.7 Å global resolution (EMD-19408). **e** CDC25A (green) catalytic

domain binds across the N- and C-terminal lobes of CDK2 (purple), whilst the C-terminal helix (indicated by the red arrow) binds at the CDK2-cyclin A (salmon) interface. N- and C-terminal cyclin box folds are indicated by N- and C-CBF respectively. **f** The local resolution within the complex varies from 2.4 to 3.0 Å (FSC 0.143) (map shown in same orientations as **e**). **g** Schematic of the domain organisation of CDC25A; the catalytic domain and C-terminal tail were expressed in this study with mutation of catalytic Cys341 to serine. Phosphorylation sites that mediate the interaction with 14-3-3 proteins are labelled. For clarity, the 16 reported phosphorylation sites within the N-terminal regulatory domain (listed in Uniprot entry P30304 and annotated in Sur et al.[36]) are omitted.

## Table 1 | Sequence conservation of selected residues that mediate trimeric complex formation

| CDK2 | CDK1 | CDK3 | CDK4 | CDK6 |
|---|---|---|---|---|
| Glu12 | Glu | Glu | Val | Glu |
| Glu42 | Glu | Glu | Gly | Glu |
| Gln131 | Gln | Gln | Glu | Gln |
| Glu162 | Glu | Glu | Val | Val |
| Asp206 | Asp | Asp | Asn | Ser |
| Glu208 | Glu | Glu | Glu | Asp |
| Asp210 | Asp | Asp | Asp | Asp |
| Phe213 | Phe | Phe | Gly | Gly |
| Arg217 | Arg | Arg | Asp | Asp |
| 9 residues | 9 conserved | 9 conserved | 2 conserved | 4 conserved |
| cyclin A (bovine) | cyclin A (human) | cyclin B (human) | cyclin E (human) | cyclin D1 (human) | cyclin D3 (human) |
| Tyr269 | Tyr | Tyr | Tyr | Ile | Thr |
| Glu272 | Glu | Glu | Lys | Thr | Thr |
| 2 residues | 2 conserved | 2 conserved | 1 conserved | 0 conserved | 0 conserved |
| CDC25A | CDC25B | CDC25C | | | |
| Lys353 | Lys | Lys | | | |
| Gln355 | Gln | Gln | | | |
| Ser435 | Ser | Ser | | | |
| Arg446 | Arg | Arg | | | |
| Glu447 | Glu | Glu | | | |
| Arg450 | Arg | Arg | | | |
| Tyr455 | Tyr | Tyr | | | |
| Arg506 | Arg | Lys | | | |
| Ser513 | Ser | Gln | | | |
| Lys514 | Arg | Leu | | | |
| 10 residues | 9 conserved | 7 conserved | | | |

Selected residues of (a) CDK2 (b) cyclin A and (c) CDC25A identified from the cryo-EM structure that mediate complex formation. The number of conserved residues across relevant CDKs, cyclins and CDC25 isoforms is indicated. Complete sequence alignments are provided in Supplementary Fig. 7.

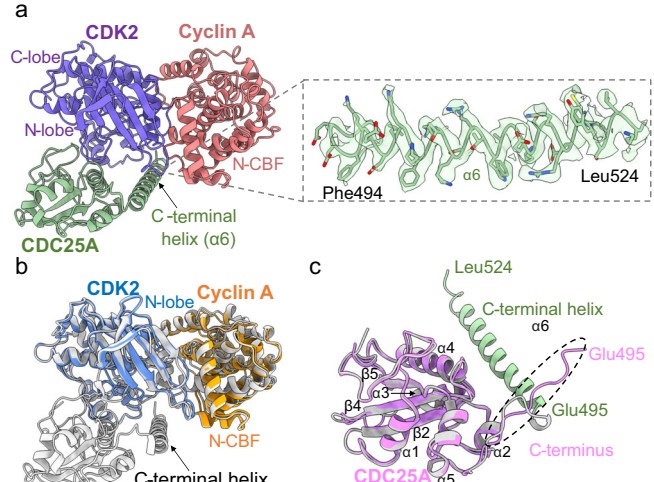

**Fig. 2 | Model of the CDK2-cyclin A-CDC25A complex. a** Structure of the CDK2-cyclin A-CDC25A complex (PDB 8ROZ) in the same orientation and colour depiction as Fig. 1e, showing overall quaternary architecture and binding of CDC25A. Zoomed panel depicts the CDC25A C-terminal helix (EM density threshold 0.248) which binds at the CDK2-cyclin A interface. Superposition of CDK2-cyclin A-CDC25A (grey) with existing (**b**) CDK2-cyclin A crystal structure (PDB 1JST[58], CDK2 in blue and cyclin A depicted in orange) and (**c**) CDC25A crystal structure (PDB 1C25[37], pink). The helical secondary structure of the CDC25A C-terminal domain observed in the cryo-EM structure, spanning residues Glu495-Leu524 (shown in green), is not observed in the CDC25A crystal structure which terminates at Glu495 (labelled in **c**).

behaviour of this C-terminal region. This flexibility is supported by evaluation of the refined atomic displacement parameters of atoms within the helix, which reveals the temperature factors across the helix are high relative to the rest of the structure (with the exception of residues 512-517 that are directly involved in contacts to CDK2 and cyclin A). Therefore, it may not be possible to detect binding on the timescales measured by HDX-MS.

### CDC25A-CDK2 interactions: CDC25A substrate recognition
Substrate recognition by CDC25A is driven by the CDK subunit, and this interaction is mediated through three distinct CDK2 regions; the G-loop (connecting the β1 and β2 strands of the N-terminal lobe, residues 9-18 containing pTyr15), the activation segment and the GDSEID-αG helix motif. Across the CDK2-CDC25A interface, multiple hydrogen bonds and salt bridges were identified by PISA (protein interfaces, surfaces and assemblies) analysis[59].

Within the CDC25A active site, clear density for CDK2 pTyr15 was observed (Fig. 3a–c). Although density for the surrounding CDC25A residues was fragmented so that sidechain locations were more ambiguous, the backbone structure for all residues of the G-loop could be modelled. In dimeric pT160CDK2-cyclin A (PDB 1JST), the G-loop is pulled towards the surface of CDK2, placing Tyr15 within hydrogen bonding distance of Glu51, and underneath the triphosphate of the ATP substrate. In the dimeric pY15pT160CDK2-cyclin A structure

(PDB 2CJM[60]) the Tyr15 phosphate group is solvent exposed and is coordinated through a network of waters to Ser46 and Thr47 at the start of the αC-helix[60]. When in complex with CDC25A, this G-loop is drawn away from the surface of CDK2, positioning pTyr15 into the shallow active site of CDC25A (Fig. 3a, b). In this extended loop conformation, pTyr15 hydrogen bonds with the mutated catalytic Ser431 residue and surrounding Glu432, Ser435, Glu436 and Arg437 residues of CDC25A (Fig. 3c).

In contrast, the CDK2 activation segment (containing pThr160) remains relatively unperturbed by CDC25A binding, adopting a conformation almost identical to that reported in the pT160CDK2-cyclin A crystal structure (PDB 1JST). Importantly, this conformation facilitates interaction with the CDC25A C-terminal helix (Fig. 3d–f). Unambiguous density for pThr160 shows the phosphate group resides in a small, positively charged pocket within the CDK2 activation segment and hydrogen bonds with surrounding CDK2 Arg126 and Arg150 residues. Analysis of the electrostatic surface potential suggests the activation segment provides a negatively charged surface that binds the opposingly charged surface of the CDC25A C-terminal helix (Fig. 3e). A significant interaction in this region is a hydrogen bond formed between CDK2 Glu162 and CDC25A Ser513, whilst neighbouring CDK2 His161 contacts CDC25A Arg520 (Fig. 3f). The cyclin A Tyr269 hydroxyl group (equivalent to human cyclin A Tyr271) is also within hydrogen bonding distance of CDC25A Ser513 and is placed to engage in a π stacking interaction of its aromatic ring with the aliphatic sidechain of CDC25A Lys514 within the C-terminal tail (see below). Tyr269 and Ile268 further create a notable hydrophobic patch on the cyclin A surface. Collectively, these interactions with the CDC25A C-terminal helix provide a mutual binding site for CDK2 and cyclin A within the trimeric complex.

Within the CDK C-terminal lobe, the CDK family has a loop linking the αF and αG helices that contains the GDSEID sequence that is conserved in CDKs 1-3. The importance of this motif to CDK1 function was first recognised through the isolation of *Schizosaccharomyces pombe cdc2* cell cycle mutants (compiled in Endicott et al.[61]). The αG

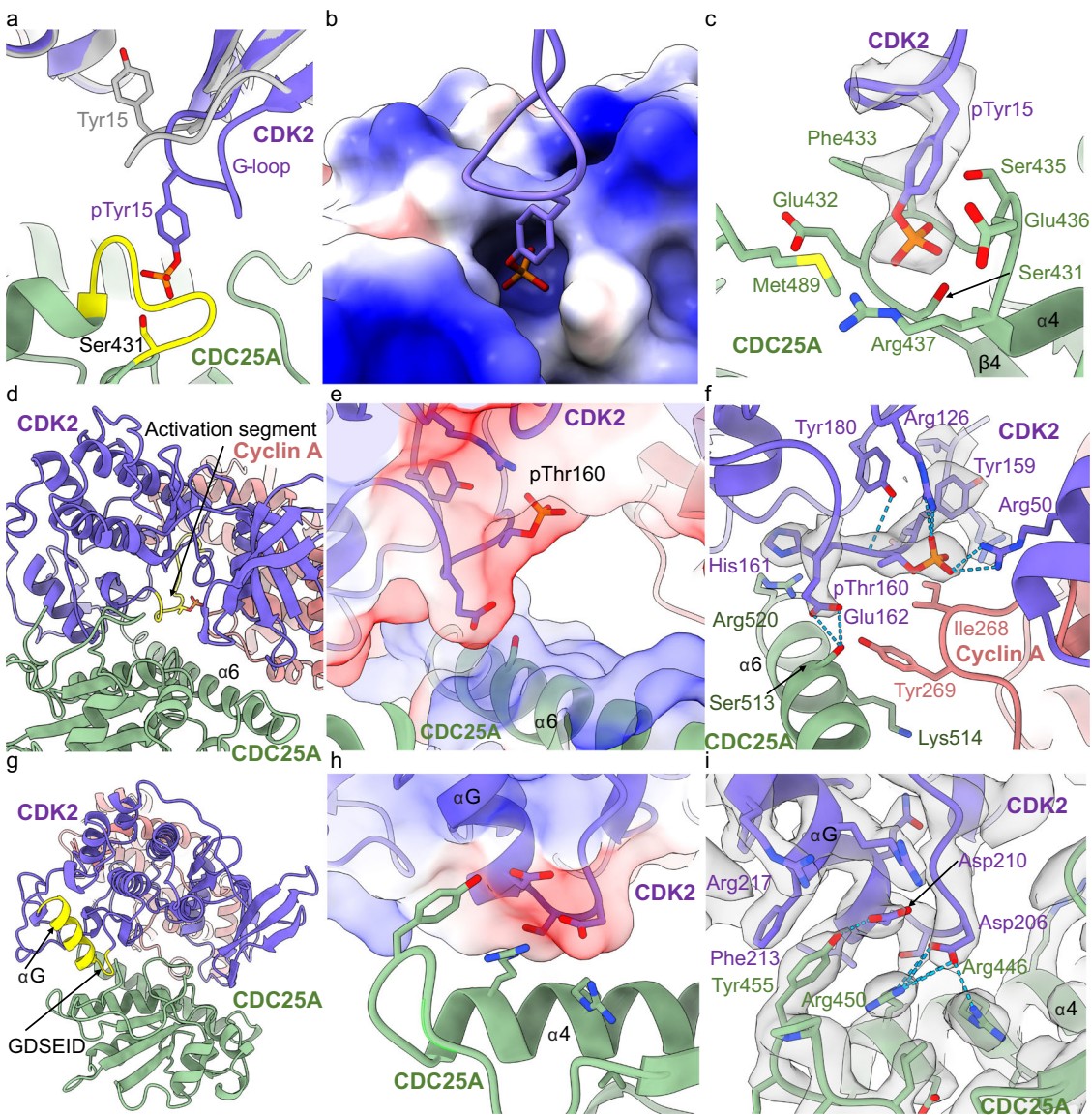

**Fig. 3 | Interactions of CDC25A with CDK2. a–c** Interactions of the CDK2 G-loop with the CDC25A active site. **a** Overlay of CDK2 (purple) with pT160CDK2-cyclin A crystal structure (PDB 1JST, grey), highlighting a change in G-loop conformation. The CDC25A CX₅R loop is highlighted in yellow. **b** Electrostatic surface of the CDC25A active site, showing binding of CDK2 pTyr15. **c** Ribbon and density diagram (0.06 threshold) for the binding of CDK2 pTyr15 in the CDC25A active site. pTyr15 and the mutated catalytic cysteine residue (Cys431Ser) are labelled in **a** and **c**. **d**–**f** Interactions of the CDK2 activation segment with the CDC25A C-terminal helix (α6). **d** Ribbon diagram of the trimeric complex, highlighting the CDK2 activation segment (yellow). **e** Electrostatic surface of the CDK2 activation segment and CDC25A C-terminal helix. **f** Ribbon and density diagram (threshold 0.17) depicting binding of the CDK2 activation segment with the CDC25A C-terminal helix, which also binds cyclin A Tyr269 (salmon). **g**–**i** Interactions of the GDSEID motif with CDC25A. **g** Ribbon diagram highlighting the CDK2 GDSEID motif and αG helix (yellow). **h** Electrostatic profile of CDK2 GDSEID motif and (**i**) Ribbon and density diagram (threshold 0.14) for binding of the CDK2 GDSEID motif (purple) with the central helix of CDC25A (green).

helix and preceding GDSEID sequence protrude over the CDC25A helix that lies C-terminal to the mutated catalytic Cys431Ser residue. This generates an interface where the GDSEID motif provides a negatively charged surface which binds the positively charged CDC25A central helix, α4 (Fig. 3h). At this interface, CDK2 Asp206 (the first Asp of the GDSEID sequence) hydrogen bonds to CDC25A Arg450 and forms a salt bridge with CDC25A Arg446 (Fig. 3i), whilst the backbone of Glu208 hydrogen bonds with the side chain of CDC25A Glu447. Additionally, Asp210 (the second Asp of the GDSEID motif) hydrogen bonds to nearby CDC25A Tyr455, which itself sits within hydrogen bonding distance to CDK2 Arg217 further along αG. In this region, the aromatic side chain of CDC25A Tyr455 also forms a π-stacking interaction with CDK2 Phe213 (Fig. 3i), which binds in a hydrophobic pocket

contributed by CDK2 residues Ile209, Phe216, Phe240, Pro241 and Trp243. Combined, these residues of the αG helix and GDSEID sequence (Asp206, Glu208, Asp210, Phe213 and Arg217) create interactions that pin the CDC25A catalytic core to the C-terminal domain of CDK2.

Additional regions of CDC25A-CDK2 interaction include an extended CDC25A loop (residues 350-358, prior to α1) that binds in the wide groove between the CDK2 N-terminal β-sheet and the C-terminal L12-αF-L13 sequence (prior to the GDSEID motif). This interaction bridges the N- and C-terminal lobes of CDK2, where CDK2 Lys129, Gln131, Thr165, Trp167, Tyr168 create a pocket that binds CDC25A Gln355. Specifically, CDK2 Gln131 hydrogen bonds with CDC25A Gln355, whilst CDC25A Lys353 hydrogen bonds to CDK2 Lys88

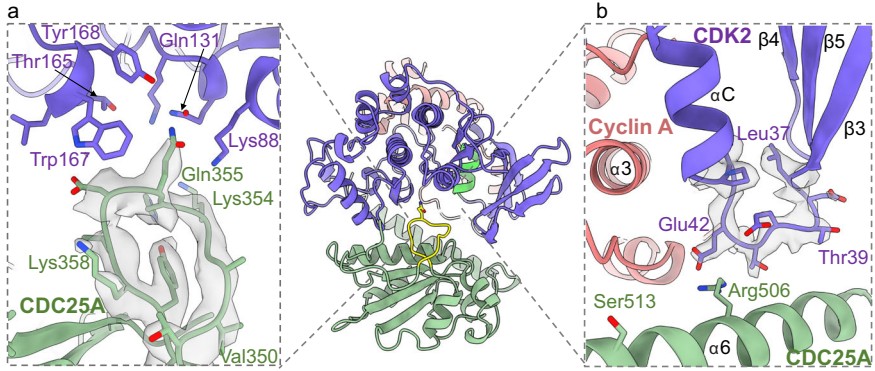

**Fig. 4 | Additional CDK2-CDC25A interactions.** Ribbon and density diagram (threshold 0.10) depicting (**a**) the binding of the CDC25A loop (residues 350–358 green) in the groove between the N- and C-terminal lobes of CDK2 (purple) and (**b**) binding of the CDK2 loop (residues 37–45, between PSTAIRE (αC) helix and β3, purple) with the C-terminal helix (α6) of CDC25A (green).

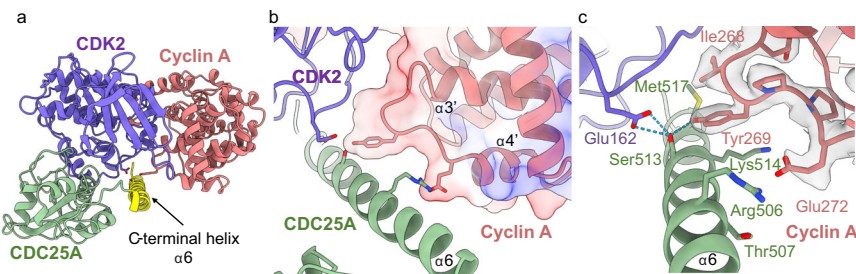

**Fig. 5 | Interactions of the CDC25A C-terminal helix with cyclin A. a** Ribbon diagram of the trimeric complex (CDK2 purple, cyclin A salmon, CDC25A green) revealing the binding of the CDC25A C-terminal helix (highlighted in yellow) at the CDK2-cyclin A interface. **b** Electrostatic surface profile of cyclin A C-CBF, highlighting the loop between α3' and α4' which interacts with the CDC25A C-terminal helix. **c** Ribbon and density diagram (threshold 0.17) depicting the binding of the CDC25A C-terminal helix (green) with cyclin A C-CBF. CDC25A Ser513 binds both cyclin A Tyr269 and CDK2 Glu162.

(Fig. 4a). Other interacting CDK2 residues are located on the strand that connects the N-terminal β-sheet (β3) to the PSTAIRE (αC) helix (Fig. 4b). This strand forms a generally negatively charged environment that binds the positively charged CDC25A C-terminal helix (α6). Most notably, CDK2 Thr39 forms a hydrogen bond with CDC25 Arg502 and Glu42 forms a salt bridge with CDC25A Arg506.

### Investigating CDC25A-cyclin A interactions

In contrast to the numerous interactions with CDK2, CDC25A makes relatively few contacts with cyclin A; only 2 hydrogen bonds and 1 salt bridge were identified between the two proteins, all of which occur across the CDC25A C-terminal helix (Fig. 5a–c). At the end of the central helix (α3') in the cyclin A C-CBF, Tyr269 (Tyr271 in the human cyclin A2 sequence) creates a π-aliphatic side chain interaction with CDC25A Lys514, whilst hydrogen bonding to CDC25A Met517 and Ser513 (which also binds the activation segment of CDK2 (Fig. 5b, c)). Further N-terminal in the loop connecting the α3' and α4' helices, bovine cyclin A Glu272 (equivalent to human cyclin A2 Glu274) forms a salt bridge with CDC25A Arg506 (Fig. 5c), a residue that also interacts with CDK2 through the PSTAIRE (αC) helix. Therefore, CDC25A Arg506, Ser513 and Lys514 play pivotal roles in securing the binding of the C-terminal helix to both cyclin A and CDK2. Moreover, the CDK2-cyclin A interface forms a generally negatively charged groove that binds the opposingly charged CDC25A C-terminal helix (Fig. 5b).

To explore the requirement for cyclin A for complex formation, we employed surface plasmon resonance to compare the affinity of CDC25A(C431S) for monomeric pY15pT160CDK2, pT160CDK2 and for pT160CDK2-cyclin A. Despite making few interactions with cyclin A, the affinity of CDC25A(C431S) for pT160CDK2-cyclin A (K$_d$ = 15 ± 1.2 μM) is significantly greater than that for monomeric pY15pT160CDK2 and pT160CDK2 (for which K$_d$ values could not be reliably determined, Supplementary Fig. 6a–c). The structure of monomeric pT160CDK2 (PDB 1B39[62]) is predicted to be incompatible with CDC25A binding because the CDK2 activation loop is folded to partially occlude the G-loop. The quality of the electron density map in this region of the monomeric pT160CDK2 crystal structure suggests this sequence is flexible and may adopt several different mobile states, whilst CDK2 residues 36-43 that precede the C-helix are also disordered and could not be modelled. We conclude that cyclin A has a significant role in facilitating the formation of the CDK2-cyclin A-CDC25A complex by enforcing changes in CDK2 conformation around the active site to make CDK2 a CDC25A substrate.

### The CDC25A C-terminal helix is required for complex formation

Despite making few interactions with CDC25A, we confirmed the requirement for the CDC25A C-terminal tail for binding to CDK2-cyclin A by assaying a set of CDC25A C-terminal tail mutants through homogenous time-resolved fluorescence (HTRF) (Supplementary Fig. 6d). By HTRF, the affinity of CDC25A(C431S) for pT160CDK2-cyclin A was 0.11 ± 0.02 μM. As predicted from the interactions described above, terminating the CDC25A construct at Glu495, and charge reversal mutations in the CDC25A C-terminal helix (R502E/K504E/R506E and K514E/R520E) abrogated binding, highlighting the importance of the C-terminal helix for trimeric complex formation.

The C-terminal sequence has previously been shown to mediate the interaction of CDC25A with 14-3-3 proteins. 14-3-3 proteins recognise CDC25A phosphorylated on Thr507 as one of two phosphoamino acids required for their bidentate interaction with ligands[40,42]. In our cryo-EM structure, Thr507 points out into solution, with no apparent CDK2-cyclin A interactions (Fig. 5c). As expected,

analysis by HTRF revealed mutation of Thr507 to a glutamate (as a phosphothreonine mimetic) had little effect on the affinity of CDC25A for CDK2-cyclin A (Supplementary Fig. 6d). To be a 14-3-3 ligand, the CDC25A tail is most likely flexible in solution to promote first recognition by the CHK1 catalytic site and then subsequent binding to the extended 14-3-3 phospho-peptide recognition cleft. This function of CDC25A further supports our hypothesis that the C-terminal region of CDC25A must be structurally dynamic but forms helical secondary structure on binding to the CDK2-cyclin A substrate.

## The CDK2 GDSEID sequence mediates CDK2-protein interactions

With the determination of the structure of CDK2-cyclin A-CDC25A, there are now structures for three protein complexes that exploit the CDK2-C-terminal lobe for complex formation. The structures of CDK2 bound to cyclin-dependent kinase subunit (CKS)1 (PDB 1BUH[50]) and kinase associated phosphatase (KAP) (PDB 1FQ1[49]) were the first to illustrate the importance of the GDSEID motif to CDK1/2 regulation.

**Comparison with the pT160CDK2-KAP complex.** KAP is a dual specificity phosphatase that dephosphorylates pThr160 within the CDK2 activation segment. Superposition of the crystal structure of KAP bound to pT160CDK2 (PDB 1FQ1) with our ternary pY15pT160CDK2-cyclin A-CDC25A complex, reveals that although the two phosphatase binding sites overlap, there are significant differences between the interfaces.

As described above, CDC25A binds across the N- and C-terminal lobes of CDK2, whereas KAP binds almost exclusively to the C-terminal lobe (Fig. 6a). In complex with KAP, the CDK2 activation segment

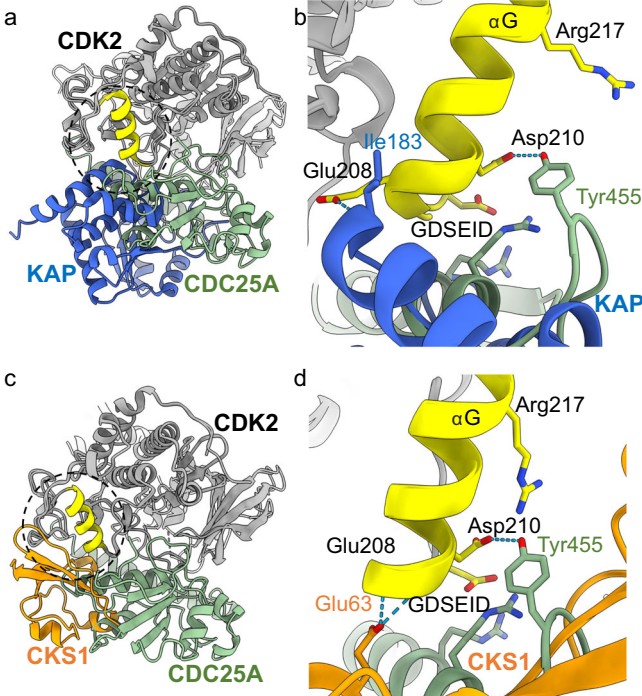

adopts an extended conformation so that pThr160 can reach the KAP active site. However, in complex with CDC25A, the activation segment sits closer to the CDK2 surface. Therefore, CDC25A and KAP appear to require and/or induce different activation segment conformations. The two phosphatases also induce different G-loop conformations. When bound to CDC25A, this loop extends away from the CDK2 surface so that pTyr15 is placed in the CDC25A catalytic site; whereas bound to KAP it adopts a more compact conformation, where the only notable interaction occurs between KAP Lys54 and CDK2 Tyr15.

The GDSEID, αG helix and the DYK motifs in the L14 loop of CDK2 (residues 235-237) contribute affinity and specificity to the CDK2-KAP interaction[49]. The inner face of the αG helix and GDSEID sequence creates a non-polar pocket that binds residues from the C-terminal helix of KAP, where CDK2 Glu208 hydrogen bonds to KAP Ile183 (Fig. 6b). In complex with CDC25A, the outer face of the αG helix provides a negatively charged surface that binds a positively charged CDC25A core helix through hydrogen bonding interactions with CDK2 Asp206, Asp210 and Arg217 (Figs. 3i and 6b). Therefore, although the GDSEID-αG helix motif provides a mutual binding site, CDC25A and KAP bind to opposite faces of the αG helix and interact with different CDK2 residues.

**Comparison with the pThr160CDK2-CKS1 complex.** CKS1 is an essential CDK accessory protein that in all eukaryotes promotes the multisite phosphorylation of CDK1 substrates by providing a phospho-amino acid binding site in trans[63]. In higher eukaryotes it is also an essential adaptor protein that regulates p27KIP1 abundance by facilitating its interaction with the SCF-SKP2 E3 ligase for ubiquitin-mediated proteasomal degradation[64,65]. The CDK GDSEID motif is also required for CKS1 binding.

Whilst the recognition sites for CDC25A and CKS1 (PDB 1BUH) partially overlap at the GDSEID region, the exact nature of the interactions is different. The binding of CKS1 to CDK2 is more comparable to that of KAP binding to CDK2. Specifically, one loop of CKS1 (residues 58–64) protrudes into a groove formed by CDK2 residues Ile209, Phe213, Phe240, Pro241, and Trp243, whilst the anti-parallel β-sheet core of CKS1 packs against the GDSEID-αG helix (Fig. 6c). This forms an extensive hydrophobic interface that spans the length of the αG helix, in which CKS1 Glu63 and Ile59 hydrogen bond with CDK2 Glu208 and Lys237 in the L14 loop that follows αG (Fig. 6d). In contrast, CDC25A interacts predominantly with the GDSEID sequence and the start of αG, forming a smaller, more hydrophilic interface strengthened by hydrogen bonds to CDK2 Asp206, Asp210 and Arg217. Therefore, the binding of CKS1 extends further across the GDSEID-αG helix and the interactions appear more hydrophobic.

CKS1 is important for CDK1 and CDK2 substrate recognition. When bound to CDK2 within a pentameric CDK2-cyclin A-CKS1-SKP1-SKP2 complex, CKS1 recognizes the phosphoThr187 of p27KIP1 which is then recruited to the SCF^SKP2 E3 ubiquitin ligase complex[64–66]. The binding site for CKS1 on CDK2 is also conserved in CDK1[50,67], and CKS1 promotes CDK1-cyclin A/B multi-site phosphorylation of substrates that include CDC25, Wee1 kinase and Myt1 kinase[63,68]. From the structural comparison, it can be hypothesised that CKS1 binding to CDK1 would impact the reciprocal regulation of CDK1 and CDC25. Association of CDK1 with CKS1 would preclude subsequent regulation of CDK1-cyclin A/B by active site phosphorylation by blocking CDC25 binding (and possibly Wee1 kinase/Myt1 kinase binding) while potentially promoting the multisite phosphorylation of CDC25 within its N-terminal sequence. A comparison of the CDK1/2-CKS1 and CDK1/2-CDC25A interfaces provides an opportunity to identify CDK mutations that might distinguish CKS1 and CDC25A binding to explore their mutual regulation of CDK1 and CDK2 activities.

In summary, it is evident from a comparison of CDK2 bound to KAP, CKS1 and CDC25A that the CDK2 GDSEID motif is an important mutual binding region for these proteins which confers selectivity by

**Fig. 6 | Comparison of KAP/CKS1/CDC25A with GDSEID binding region of CDK2. a** Overlay of the trimeric complex (CDK2-Cyclin A in grey, CDC25A in green) with the CDK2-KAP crystal structure (PDB 1FQ1, KAP in blue) and (**c**) CDK2-CKS1 crystal structure (PDB 1BUH, CKS1 in orange). The CDK2 GDSEID sequence and αG helix are highlighted in yellow. **b** Ribbon diagram of the GDSEID region of CDK2 bound to CDC25A (green) and KAP (blue), which bind to opposing faces of the αG helix. **d** Ribbon diagram of the GDSEID region of CDK2 bound to CDC25A (green) and CKS1 (orange), showing an overlap in the binding region but differences in specific residue interactions.

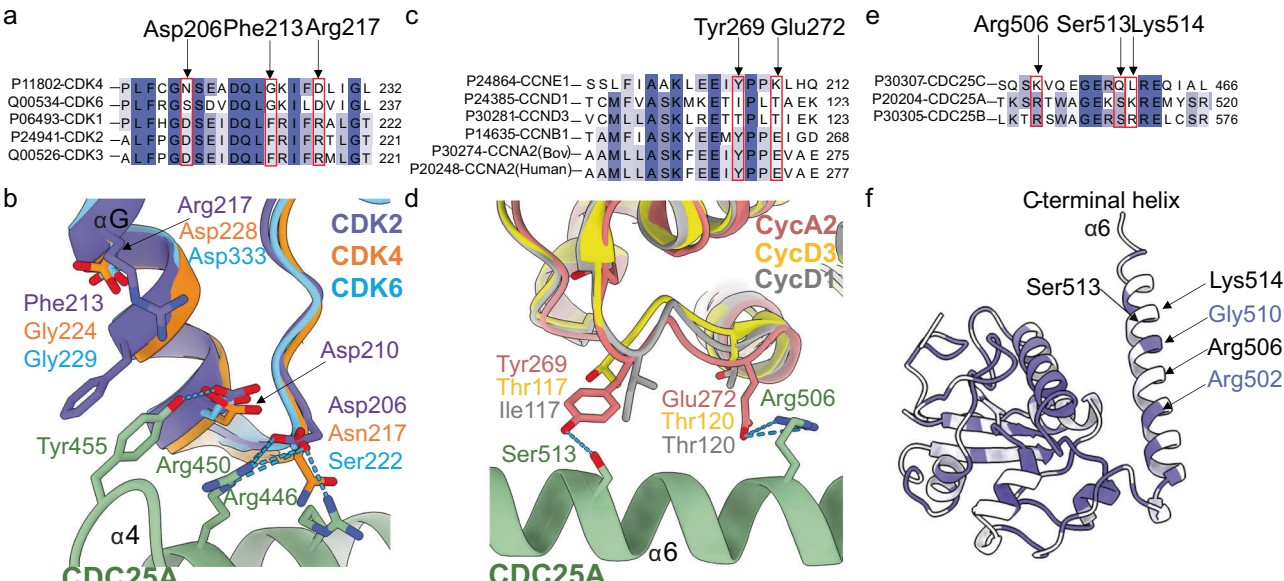

**Fig. 7 | Sequence conservation at CDC25A binding regions.** Sequence conservation is indicated in purple (darker shade = identical, lighter shade = similar, white = different) and selected residues directly involved in CDC25A binding are highlighted in red boxes. **a** Sequence conservation of CDK2 vs CDK1, 3, 4 and 6 around the GDSEID-αG helix motif. **b** Structural overlay of CDK2 (purple) with CDK4 (orange) and CDK6 (blue) showing changes in residue identity for Asp206, Phe213 and Arg217 at the GDSEID-αG helix motif (**c**) Sequence conservation of cyclin A vs cyclin B, E, D1 and D3 around the central helix (α3') in the cyclin A C-CBF. **d** Structural overlay of cyclin A (salmon) with cyclin D1 (grey) and cyclin D3 (yellow), highlighting the changes in residues identity for Tyr269 and Gly272 in the C-CBF. **e** Sequence conservation of CDC25A vs CDC25 B and C at the C-terminal helix region. **f** Ribbon diagram of CDC25A coloured by sequence conservation. CDC25A residues Arg506, Ser513 and Lys514 highlighted on the C-terminal (α6) helix are non-conserved in CDC25C.

engaging different residues and hydrophobicity surface profiles. That it is a hotspot for protein binding provides the mechanistic explanation for the identification of yeast cell cycle mutants within the sequence[69,70]. Further affinity and selectivity for CDC25A is provided by the N-terminal CDK2 binding regions, notably the G-loop and PSTAIRE helix. These regions appear less important for the binding of KAP and are irrelevant for CKS1 binding.

**Sequence alignments to identify CDC25 substrate preferences**

This cryo-EM complex provides a template from which to assess the sequence differences between members of the CDK, cyclin and CDC25 families at key residues that mediate the interaction between CDC25A and CDK2-cyclin A to identify potential partner preferences (Table 1, Supplementary Fig. 7). In addition to CDKs 1, 2 and 3, CDK4 has been reported to be phosphorylated on the conserved tyrosine within the G-loop[71].

Sequence alignment of CDK2 with CDKs 1, 3, 4 and 6 (Table 1, Supplementary Fig. 7a, b) revealed that CDK1 and CDK3 are conserved at all 9 selected residues that play critical roles in mediating the CDK2-CDC25A interaction. In contrast, only 2 and 4 residues are conserved in CDK4 and CDK6 respectively (Table 1). Both CDK4 and CDK6 lack the GDSEID sequence (GNSEAD in CDK4 and GSSDVD in CDK6), and as the most prominent C-terminal interaction site, the sequence changes of CDK2 Asp206, Phe213 and Arg217 to asparagine, glycine and aspartic acid in CDK4 and to serine, glycine and aspartic acid in CDK6 would be expected to significantly impact their interaction with CDC25A (Fig. 7a, b).

Prior to the activation segment, CDK2 Gln131 is required to stabilise the extended CDC25A loop that binds in the wide groove between the N- and C-terminal lobes of CDK2. This residue is conserved in CDK6 but not in CDK4, in which Gln131 is a glutamate. Within the activation segment, Glu162 hydrogen bonds to the CDC25A C-terminal helix but the inclusion of a valine at this position in the CDK4 and CDK6 sequences precludes this interaction. CDK4 and CDK6 also differ in sequence at key interacting residues within the N-terminal lobe. Whereas CDK6 conserves CDK2 N-terminal residues Glu12 and Glu42 (required to bind to the CDC25A active site and C-terminal helix respectively) they are mutated to valine and glycine respectively in CDK4.

Conservation analysis of cyclin A against B, E, and D-type cyclins at CDC25A recruitment sites (Table 1b, Supplementary Fig. 7c, d) is somewhat simplified given the relatively limited binding profile observed between CDC25A and cyclin A (Fig. 5, Table 1). The key bovine cyclin A2 Tyr269 residue is conserved in cyclins B and E, but as a threonine in the D-type cyclins its ability to hydrogen bond to Ser513 on the CDC25A C-terminal helix is lost (Fig. 7c, d). Cyclin A Glu272, which forms a salt bridge with Arg506 on the CDC25A C-terminal helix, is also conserved in cyclin B, but changes to a lysine in cyclin E and to an isoleucine and threonine in cyclins D1 and D3 respectively. These residue changes in the D-type cyclins indicate binding of CDC25A may be compromised.

Ten CDC25A residues make significant interactions with CDK2-cyclin A, of which 9 and 7 are conserved in CDC25B and CDC25C respectively (Table 1, Supplementary Fig. 7e, f). CDC25A Arg446, Glu447, Arg450 and Tyr455 are key in promoting interaction with the CDK2 GDSEID motif and αG helix (Fig. 3i); these residues are conserved across CDC25B and C, as are CDC25A residues Ser435, Arg437 and Lys353 which bind the CDK2 G-loop (Fig. 3c). However, the CDC25A C-terminal helix residues that are vital for mediating interaction with both CDK2 and cyclin A (Fig. 5) have conservative changes in CDC25B; Arg506 and Ser513 are conserved and Lys514 changes to an arginine, but are non-conserved in CDC25C, in which they are Lys452, Gln459 and Leu460 respectively (Fig. 7e, f). Combined these residue changes could significantly impact the engagement of the C-terminal helix of CDC25C to CDK2-cyclin A.

**AlphaFold Multimer models to probe CDC25A substrate preferences**

This sequence conservation analysis of critical residues involved in CDC25A recruitment suggests that CDK1/2-cyclin A, CDK1-cyclin B and

CDK2/3-cyclin E complexes are suitable binding partners for CDC25A. This is supportive of the notion that CDC25A aids in both G1/S and G2/M transitions. However, CDK4-cyclin D complexes appear to be unlikely substrates for CDC25A based on sequence conservation. To further test this hypothesis, we employed AlphaFold Multimer[72] to assess the potential for CDK4-cyclin D3 to bind to CDC25A (Supplementary Fig. 8).

To benchmark our AlphaFold Multimer analysis, we first determined its ability to predict the structure of CDK2-cyclin A bound to CDC25A (Supplementary Fig. 8a). AlphaFold Multimer predicted this structure with significant accuracy: the five highest ranked models all indicated that CDC25A binds predominantly to CDK2, but they varied in the structural prediction for the CDC25A C-terminal tail. Notably, model 2 aligned to the cryo-EM determined structure with an overall RMSD of 0.8 Å.

We next modelled the structures of CDK4-cyclin D3-CDC25A (Supplementary Fig. 8b). The five highest ranked models consistently modelled CDC25A as binding to the cyclin subunit with no predicted interaction with CDK4. As illustrated by the Predicted Aligned Error (PAE) plots, confidence in each model was low. Thus, the AlphaFold Multimer models agree with the comparative sequence analysis and suggest that CDK4 is not a CDC25A substrate.

We hypothesise that where phosphorylation of CDK4 Tyr17 occurs, it may be dephosphorylated by other means. The detection of interactions between CDK4-cyclin D and CDC25 may result from CDK4 recognizing CDC25 proteins as substrates mediated through recruitment motifs within the CDC25 N-terminal sequence[39]. The determination of the residues that mediate the interaction between the CDC25A catalytic domain and CDK2-cyclin A now provides a structural basis for further experiments to establish the importance of CDK4 inhibitory phosphorylation to regulation of its activity and the requirement for CDC25 phosphatases to modulate this post-translation modification.

As key regulators of cell cycle transitions and checkpoint pathways, CDC25A and CDC25B are frequently overexpressed in human cancers and often associated with poor prognosis. Consequently, the CDC25 isoforms are attractive yet challenging anticancer drug targets. Previous docking and molecular dynamic simulations have proved useful in predicting the overall binding mode of CDC25B[73], and a fragment-based study has provided proof-of-concept in targeting the CDC25-CDK interaction[74]. We envisage this cryo-EM structure will facilitate the identification of allosteric sites that may be amenable to the development of inhibitors that prevent complex assembly or disrupt CDC25A catalysis.

## Methods

All vector constructs for expression of the proteins used in this study are available from the corresponding author on request.

### Protein production, purification and complex formation

Human CDK2 phosphorylated on Tyr15 and Thr160, or only on Thr160, bovine cyclin A2, and human CDC25A and CDC25A mutant proteins were all expressed in recombinant *E. coli* cells and subsequently purified by affinity, ion-exchange and size exclusion chromatography steps. Detailed protocols for the various protein purifications are provided in the Supplementary Information.

### Formation of the CDK2-cyclin A-CDC25A complex

To generate the trimeric complex, purified pY15pT160CDK2, cyclin A and CDC25A were mixed in a molar ratio of 1:1.5:2 and incubated on ice for 1 hr. The complex was separated from monomeric and dimeric species by analytical gel filtration using an Akta Pure Microsystem (Cytiva) with a Superdex S200 10/300 column equilibrated in HBS (40 mM HEPES, 200 mM NaCl, 1 mM DTT, pH 7.4). Fractions containing the trimeric complex, as confirmed by SDS-PAGE, eluted from the column at a concentration suitable for cryo-EM studies (0.5–2.0 mg mL$^{-1}$) without the need for concentration. The complex was snap-frozen in liquid nitrogen before application to cryo-EM grids.

### Cryo-EM studies

All cryo-EM data were processed in CryoSPARC[75] unless otherwise stated.

**Initial sample preparation, data collection and processing.** Preliminary cryo-EM samples were prepared using an FEI Vitribot mark IV set at 80% humidity and 4 °C. 2.5 μL aliquots of the trimeric complex at -0.3 mg mL$^{-1}$ were applied to QuantiFoil R2/2 200 mesh grids (EM Resolutions (QR22200Cu100)) that had been glow discharged using a PELCO easiGlow™ Glow Discharge System for 1 min at 20 mAmp/0.26 mBar using ambient atmosphere. Grids were blotted with a blot force −10, blot time of 2–6 s and wait time of 3 s before plunge-freezing in liquid ethane.

Grids were screened and initial data collected on a 200 kV Glacios microscope (Thermo Fisher Scientific) with a Falcon 4 direct electron detector, housed at the York Structural Biology Laboratory (University of York, UK). Cryo-EM images were acquired at an accelerating voltage of 200 kV and nominal magnification of ×240,000, giving a calibrated pixel size of 0.574 Å. Data were collected using AFIS in EER format, with an exposure time of 6.5 s and total fluence of 50 e/Å$^2$. A defocus range of −0.6 to −2.0 μm every 0.2 μm was used and autofocus was performed after centring. A 100 μm objective aperture was used. A total of 11,955 movies were collected across two grids.

Movies were motion corrected using Patch Motion Correct Multi and the contrast transfer function (CTF) parameters were estimated using Patch CTF Correction Multi. Particles were picked using blob picker with a 40 Å minimum and 120 Å maximum particle diameter. Particle picks were inspected and filtered according to their power and NCC scores. Accepted particles were extracted with a box size of 384 pixels (px) and Fourier cropped to 96 px. Extracted particles were subject to multiple rounds of 2D classification using 40 iterations, a batch size of 200, and an initial classification uncertainty factor of 5. Significant preferential orientation was evident from the 2D classes. These particles were used to train Topaz picking (using the ResNet8 model architecture on a subset of 1000 micrographs[52,53]) and subsequent particle picking with the Topaz model provided additional 2D-views. Ab-initio reconstruction followed by homogenous refinement resulted in a map that encompassed the full complex, however, significant preferential orientation limited 3D refinement, leading to streaking and poor fitting to secondary structure.

**Improving sample orientation distribution with CHAPS.** To reduce the preferential orientation, samples were spiked with 0.1, 0.2, 0.5 and 1.0 X CMC (critical micelle concentration, CMC = 8 mM) CHAPS prepared in HBS buffer (40 mM HEPES, 200 mM NaCl, 1 mM DTT, pH 7.4) immediately prior to grid preparation. 2.5 μL aliquots of the trimeric complex (-1.8 mg/mL) freshly spiked with CHAPS were applied to QuantiFoil R1.2/1.3 200 mesh grids (EM Resolutions (QR1213200Cu100)) that had been glow discharged using a PELCO easiGlow™ Glow Discharge System for 1 min at 20 mAmp/0.26 mBar using ambient atmosphere. The grids were blotted with a blot force −10, blot time of 4–12 s and wait time of 3 s before plunge-freezing in liquid ethane using a FEI Vitribot mark IV set at 80% humidity and 4 °C.

Grids were screened, and initial data were collected on a Glacios microscope (Thermo Fisher Scientific) with a Falcon 4 direct electron detector (York Structural Biology Laboratory). At each CHAPS concentration, -1000 exposures were collected at an accelerating voltage of 200 kV, nominal magnification x240,000, a calibrated pixel size of 0.574 Å, exposure time of 3.2 s and a total fluence of 50 e/Å$^2$. Data collection was set up using the multigrid function in the EPU Software

(Thermo Fisher Scientific) as described above for the initial data collection.

All data sets were processed separately to 2D classification to identify the CHAPS concentration most beneficial for improving the particle orientation distribution. Movies were motion corrected using Patch Motion Correct Multi and the contrast transfer function (CTF) parameters estimated using Patch CTF Correction multi. Particles were picked using blob picker with a 40 Å minimum and 120 Å maximum particle diameter. Particle picks were inspected, filtered according to their power and NCC scores and extracted with a box size of 384 px, Fourier cropped to 96 px to give a pixel size of 2.296 Å. Extracted particles were subject to multiple rounds of 2D classification using 40 iterations, a batch size of 200 and an initial classification uncertainty factor of 5. The positive effect of CHAPS on the particle orientation distribution was evident at all concentrations tested, however, more unique views were identified at 0.5 and 1.0 X CMC. These grids were chosen for further data collection.

**Final high resolution data collection.** The final high resolution data set was collected across 2 grids (Quantifoil R1.2/1.3 200 mesh) prepared with 1.8 mg mL⁻¹ trimeric complex and 0.5 and 0.1 X CMC CHAPS. Cryo-EM images were acquired on a 300 kV FEI Titan Krios equipped with a BioQuantum K3 detector and imaging filter, housed at the Electron Bio-Imaging Centre (Krios 1, eBIC, Diamond Light Source, UK). Data were collected at an accelerating voltage of 300 kV and x150,000 magnification, yielding a pixel size of 0.825 Å, in counting mode with an energy filter slit width set at 20 eV. Data collection was set up using the multigrid function in the EPU Software (Thermo Fisher Scientific) using AFIS with 3 exposures per hole. Data were collected in fifty frame movies in TIFF format, with an exposure time of 1.92 s, a dose of 1.010 e/Å² per frame and a total accumulative dose of 50.49 e/Å². A defocus range of −0.6 to −2.0 μm every 0.2 μm was used and autofocus was performed after centring. A 100 μm objective aperture was used. A total of 13,242 movies were recorded.

**Data processing, and structure determination.** Movies were patch motion corrected using Patch Motion Correct Multi and the contrast transfer function (CTF) parameters estimated using Patch CTF Correction multi. Particles were picked using automated blob picker with a 40 Å minimum and 120 Å maximum particle diameter, resulting in ~6,000,000 particle picks. Particles were inspected and filtered according to their power and NCC scores to yield a particle set of 4,981,705 particles. Accepted particles were extracted with a box size of 288 px, Fourier cropped to 96 px to give an effective pixel size of 2.475 Å. Extracted particles were subjected to multiple rounds of 2D classification using 40 iterations, a batch size of 200 and an initial classification uncertainty factor of 5 (same parameters applied to all 2D classification steps unless otherwise specified). Following 2D classification, 1,231,048 particles from 76 classes were used to generate 2 ab-initio models, with a 0.01 class similarity cut-off. Class 0 of 757,792 particles was chosen as the best initial model, and particles were re-extracted with a box size of 288 px without Fourier cropping to give a pixel size of 0.825 Å. Using class 0 ab-initio as the starting model, 3D refinement was performed using the homogeneous refinement with a 10 Å maximum alignment resolution, followed by non-uniform refinement[76] (no max align) to yield a 3.2 Å (FSC 0.143) cryo-EM map of the trimeric complex.

To remove further heterogeneity, particles constituting this reconstruction were classified by 3D heterogenous refinement using the previously generated ab-initio models and a refinement box size of 144 px. Class 0 (534,740 particles, 5.6 Å) was selected as the best class, re-extracted with a box size of 288 px and refined by homogenous refinement (10 Å maximum alignment resolution) and non-uniform refinement to generate an improved reconstruction at 3.05 Å resolution (FSC 0.143).

Particles constituting this reconstruction were used to train Topaz using the ResNet8 model architecture on a subset of 1000 micrographs previously denoised with Topaz Denoise[52,53]. The resulting Topaz model was used for picking from the full set of 13,242 Topaz denoised micrographs. Following Topaz picking, 2,105,240 particles were extracted with a box size of 288 px, cropped to 144 px with an effective pixel size of 1.65 Å, and 2D classified to remove poor particles. The remaining 1,898,974 particles were sorted into 6 ab-initio models using a 0.01 class similarity cut-off, and further 3D classified and refined by heterogeneous refinement using the 6 ab-initio models. Particles from the best 3D heterogenous class (670,852 particles) were re-extracted without Fourier cropping to give an effective pixel size of 0.825 Å and refined by homogeneous refinement with a 10 Å maximum alignment resolution. The 3D reconstruction was further refined by non-uniform refinement[76] to yield a 2.9 Å reconstruction of the trimeric complex. This reconstruction was used as a reference to perform per-particle motion correction[77] in cryoSPARC, after which non-uniform refinement was repeated with the polished particle set to yield the final 2.7 Å resolution EM map. The local resolution variation was estimated via the local resolution estimation job at 0.143 FSC threshold.

**Model generation, refinement and validation.** The resolution of the 3D reconstruction was sufficient to permit automated model building using ModelAngelo[54], which was performed using a FASTA sequence containing the expressed residues of all three proteins. The model generated by ModelAngelo showed good fitting to the EM map, however, the first 26 N-terminal residues of CDK2 (containing the G-loop) and two loop regions of CDC25A (413–424 and 151–154) were modelled with low confidence. These regions could be discerned by manual modelling according to the EM map, to generate a complete model of the trimeric complex.

The completed model and EM map were imported into Phenix[56] for model refinement and validation. The fitting of the model was improved with real-space refinement followed by manual refinement in Coot[55] based on MolProbity[78] outlines. The model, cryo-EM map and map-to-model quality was assessed in Phenix using Comprehensive Validation[79] before deposition to the Protein Data Bank (PDB 8ROZ) and the Electron Microscopy Data Bank (EMD-19408) Full data collection, refinement and validation statistics are given in Supplementary Table 1.

All structural figures were generated in UCSF ChimeraX[80,81].

**Homogenous time resolved fluorescence.** HTRF assays were carried out essentially as described[82]. Mutant (Cys431Ser) GST-CDC25A (at 100 nM) and biotinylated Avi-tagged pT160CDK2-cyclin A (at an 11-point 2-fold dilution series starting at 8 μM) were prepared in HTRF buffer A (50 mM HEPES, 100 mM NaCl, 1 mM DTT and 0.1 mg mL⁻¹ BSA) and incubated together for 60 min at 4 °C. 4 nM Tb labelled anti-GST antibody (monoclonal antibody GST-Tb cryptate, (Revitty 61GSTTLB)) and streptavidin labelled with XL665 (SAXL665, (Revitty 610SAXLB) at 1/8th the concentration of the biotinylated pT160CDK2-cyclin A, were prepared in HTRF buffer B (50 mM HEPES, 100 mM NaCl, 1 mM DTT and 0.1 mg mL⁻¹ BSA) and added to each well. The plate was incubated for a further 120 min at 4 °C, before being scanned. Samples were excited using a wavelength of 337 nm and emission spectra measured at 620 nm and 665 nm (PHERAstar FS (BMG LABTECH)). Binding curves were plotted using GraphPad Prism 6 from which the $K_d$ values were determined. The curves shown are representative binding curves from at least two biological replicates each run in triplicate and carried out on separate days.

**Hydrogen-deuterium exchange mass spectrometry.** HDX-MS experiments were carried out using an automated HDX robot (LEAP Technologies, Fort Lauderdale, FL, USA) coupled to an

M-Class Acquity LC and HDX manager (Waters Ltd., Wilmslow, Manchester, UK). Proteins CDC25A(Cys431Ser), pY15pT160CDK2-cyclin A and pY15pT160CDK2·cyclin A-CDC25A(Cys431Ser)) were diluted to 10 µM in equilibration buffer (40 mM HEPES pH 7.4, 150 mM NaCl, 1 mM TCEP.HCl) prior to analysis. 5 µl sample was added to 95 µl deuterated buffer (40 mM HEPES pD 7.4 150 mM NaCl, 1 mM TCEP.HCl) and incubated at 4 °C for 0.5, 2, 10 or 30 min with each condition being acquired in triplicate along with three t = 0 conditions using protonated buffer to provide the baseline mass. Following the labelling reaction, samples were quenched by adding 75 µl of the labelled solution to 75 µl quench buffer (50 mM potassium phosphate, 0.05% DDM which was pH adjusted to give a final quenched pH of ~ 2.5). 50 µl of quenched sample were passed through a home-packed pepsin column using agarose immobilised pepsin (Thermo Fisher Scientific) at 40 µl min⁻¹ (20 °C) and a Van-Guard Pre-column Acquity UPLC BEH C18 (1.7 µm, 2.1 mm × 5 mm, Waters Ltd., Wilmslow, Manchester, UK) for 3 min in 0.3% formic acid in water. The resulting peptic peptides were transferred to a C18 column (75 µm × 150 mm, Waters Ltd., Wilmslow, Manchester, UK) and separated by gradient elution of 0–40% MeCN (0.1% v/v formic acid) in H2O (0.3% v/v formic acid) over 12 min at 40 µl min⁻¹. Trapping and gradient elution of peptides was performed at 0 °C. The HDX system was interfaced to a Synapt G2Si mass spectrometer (Waters Ltd., Wilmslow, Manchester, UK) using electrospray ionisation with a capillary voltage of 3 kV, cone gas of 50 L/h, desolvation gas 300 L/h and nebuliser of 6.5 bar. Desolvation and source temperatures were 120 and 80 °C respectively. HDMSE, dynamic range extension and sensitivity modes (Data Independent Analysis (DIA) coupled with IMS separation and single reflectron time-of-flight) were used to separate peptides prior to CID fragmentation in the transfer cell. Argon pressure in the trap and transfer cells was $2.4 \times 10^{-2}$ mbar, helium pressure was 4.21 mbar and nitrogen pressure in the drift tube was 3 mbar. Wave heights and velocities were 311 m/s and 4 V for the trap, 650 m/s and 30 V for the IMS and 175 m/s and 4 V for the transfer cell. Signals were acquired over m/z 50-2000 for 0.6 s alternating between high and low energy scans with a collision energy ramp of 18-40 V in the high energy scan to induce peptide fragmentation. HDX data were analyzed using PLGS (v3.0.2) and DynamX (v3.0.0) software supplied with the mass spectrometer. A minimal database was used for peptide ID containing the target sequences and pepsin. Restrictions for identified peptides in DynamX were as follows: minimum intensity: 2500, minimum products per MS/MS spectrum: 3, minimum products per amino acid: 0.3, maximum sequence length: 18, maximum ppm error: 10, file threshold: 4/6. Following manual curation of the data, hybrid plots were generated using Deuteros 2.0 at 0.01 significance[83]. The mass spectrometry proteomics data have been deposited to the ProteomeXchange Consortium with the dataset identifier PXD050866.

**Surface plasmon resonance.** All measurements were collected using the Biacore S200 instrument (Cytiva). A CM5 chip (Cytiva BR100530) was functionalised using the GST Capture Kit (Cytiva BR100223) that includes a polyclonal goat anti-GST antibody and Amine Coupling Kit (Cytiva BR100050) according to the manufacturer's protocols. Specifically, the CM5 chip surface was primed in running buffer and activated via NHS/EDC injection. Anti-GST antibody at 20 µg/mL was applied to the surface and ethanolamine was used to cap unreacted carboxylates. An average immobilisation level of 7000 RU was achieved. Recombinant GST provided in the GST Capture Kit was utilised to block high-affinity sites per the manufacturer's recommendation. Solutions of GST and GST-CDC25A(Cys431Ser) were prepared at 50 µg/mL, pY15pT160CDK2 at 25 µM, and pT160CDK2 and pT160CDK2-cyclin A at 30 µM. Two-fold dilutions in a volume of 100 µL were prepared over seven points for

each protein analyte with a buffer control for double referencing. A multicycle method at 20 °C was utilised in which each cycle consisted of GST capture on flow cell 1, GST-CDC25A capture on flow cell 2, sample injection in high performance mode over both flow cells, and finally an injection of regeneration solution over both flow cells. GST and GST-CDC25A were captured for 12 s at 10 µL/min then samples were injected at 20 µL/min for 120 s followed by a 300 s dissociation period. Regeneration occurred over a 12 s injection at 30 µL/min. For analysis, data were double referenced (Fc 2-1, with subtraction of preceding buffer control) in Biacore S200 Evaluation Software (Version 1.0) and exported for image generation in GraphPad Prism 6.

**AlphaFold Multimer modelling.** Alphafold2 Multimer was accessed via ColabFold[72] (v1.5.5) and used to predict the binding of CDC25A to CDK2-cyclin A and CDK4-cyclin D3 complexes. Single amino acid protein sequences for CDK2-cyclin A-CDC25A and CDK4-cyclin D3-CDC25A were inputted with ":" to specify inter-protein chain breaks for modelling complexes. AlphaFold multimer was subsequently run with default settings.

### Reporting summary

Further information on research design is available in the Nature Portfolio Reporting Summary linked to this article.

## Data availability

The cryo-EM map and model of the trimeric CDK2-cyclin A-CDC25A complex generated in this study have been deposited in the Protein Data Bank and the Electron Microscopy Data Bank with accession numbers PDB 8ROZ and EMD-19408. The mass spectrometry proteomics data have been deposited to the ProteomeXchange Consortium via the PRIDE[84] partner repository with the dataset identifier PXD050866. The Surface Plasmon Resonance and the Homogenous Time-Resolved Fluorescence data generated in this study are provided in the Supplementary Information. Source data are provided with this paper in the Source Data file. Accession codes mentioned in this study; PDB 1JST, PDB 1C25, PDB 1CWR, PDB 3OP3, PDB 2CJM, PDB 1B39, PDB 1BUH, PDB 1FQ1, PDB 8ROZ, EMD-19408, PDX050866.

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

## Acknowledgements

This research was supported by the Medical Research Council (Grant References MR/N009738/1 and MR/V029142/1, J.A.E. and M.E.M.N.). Cryogenic electron microscopy was carried out at the Electron Bio-imaging Centre (eBIC) at the Diamond Light Source under the Northern England Cryo-EM Consortium (proposal BI28576-22), and the Cryo-EM facility at the University of York (Glacios microscope funded by The Wellcome Trust grant number 206161/Z/17/Z). We wish to thank Dr D. Zabeo at the Electron Bio-imaging Centre for facilitating Cryo-EM data collections, Dr A. Baslé at Newcastle University for assistance with data management and Dr A. Calabrese at the University of Leeds for his assistance with HDX-MS data deposition. The LEAP sample handling robot used in the HDX-MS work was a kind donation from Waters UK. We thank colleagues who contributed to the early stages of this work; T. Johnson, M. Morgan, S. Platsaki, W. Stanley and K. Yata. For the purpose of open access, the authors have applied a Creative Commons Attribution (CC BY) licence to any Author Accepted Manuscript version arising from this submission.

## Author contributions

Rhianna Rowland: Conceptualization, methodology (cryo-EM sample preparation and data acquisition), investigation (protein production, cryo-EM analysis and structure determination), visualisation and writing (original draft, review and editing). Svitlana Korolchuk: methodology (protein production) and investigation (HTRF assays). Marco Salamina: Conceptualization, investigation, collaboration. Natalie Tatum: methodology (SPR assays). James Ault: Investigation (HDX data collection and analysis). Sam Hart: Resources (facilitation of cryo-EM data acquisition. Johan Turkenburg: Resources (facilitation of cryo-EM data acquisition). James Blaza: Funding acquisition, cryo-EM data acquisition. Jane Endicott: Conceptualization, resources, supervision, funding acquisition, writing (original draft, review and editing). Martin Noble: Conceptualization, resources, supervision, funding acquisition, writing (original draft, review and editing). All authors reviewed the manuscript.

## Competing interests

Some of the work in the authors' laboratory is supported by a research grant from Astex Pharmaceuticals. The authors declare no competing interests.
