## [Peer Review File · Nature Communications]

REVIEWER COMMENTS

Reviewer #1 (Remarks to the Author):

In this manuscript Rowland et al. present a cryo-EM structure of the CDK2-cyclin A-CDC25A complex. Overall, this manuscript is well written, the cryo-EM work is of high-quality and the scientific finding is of high interest.

CDC25 phosphatases regulate the cell cycle by removing inhibitory phosphorylation on the G-loop of CDK-cyclin complexes, thereby activating the CDK-cyclin complexes, which are master regulators of the cell cycle. Although the structure of CDC25 proteins on its own was already known, in this work the authors managed to obtain a snapshot of a CDK-cyclin-CDC25 complex thereby elucidating its stoichiometry, mechanisms of assembly and activation. These findings are important for the mechanistic understanding of the cell cycle, which is a process pivotal for life and deregulated in diseases. For all of these reasons I would recommend publication of this manuscript in Nature Communications. I have only a few minor comments with the hope to improve the quality of this article even further:

1) Page 4 “one of the smallest asymmetric particles to have been solved by cryo-EM”; cryo-EM structures are determined and not solved as in X-ray crystallography, therefore “solved” need replacing with “determined”.

2) Top of page 5, comparisons between the authors’ cryo-EM structure and other CDKs cyclins: I appreciate the sequence conservation analysis performed by the authors, however I am wondering if AlphaFold multimer could be used to support these predictions further. The AlphaFold predictions could also help enriching the discussion/conclusion section (last paragraph of the main text, pages 18-22).

3) Pages 6 and 7. Most of the paragraph titled “Sample preparation and preliminary data collection” could be moved in the Method section as it contains very technical details (e.g. ModelAngelo was used for...), which are interesting for the cryo-EM reader and not for the general reader. This excess of technical details in the main text may obscure the biological findings coming from the structure.

4) Page 9, comparing with PDB 1C25A, RMSD value should be indicated. In the following sentence, PTP C(X)5R loop could be depicted in some figure and Cys431Ser and pTyr15 should be referenced to Figure 3. In general referencing and labelling in this paragraph should be improved.

5) Page 10, beginning of paragraph CDK2-CDC25A interactions... Similar issues with referencing and labelling as in point 4. Can the G-loop be depicted better perhaps in Figure 3? GDSEID-alphaHelix motif, Figure 3 should be referenced.

6) Page 15, bottom paragraph, again similar issue, Thr507 should be referenced to some Figure... maybe depict in Figure 5c? The result of mutation Thr to Glu should be referenced to something?

Reviewer #2 (Remarks to the Author):

Comments on Rowland et al.:

The eukaryotic cell cycle is controlled by an intricate regulatory system that ensures faithful execution of critical cellular processes in a coordinated fashion. Cyclin-dependent kinases (CDKs) are at the core of these regulatory circuits. Therefore, they are themselves subject to multiple layers of regulation, including by association with protein partners (such as cyclins) and by post-translational modifications, most notably phosphorylation and de-phosphorylation of key residues. In their manuscript, Rowland et al. describe the cryo-EM analysis of the CDK2-cyclin A complex bound to the phosphatase CDC25A, which dephosphorylates and thereby de-represses CDK2 and other cyclin-dependent kinases. The work described by Rowland et al. reveals the mechanism by which CDC25A recognises CDK2-cyclin A. Sequence comparisons suggest which additional substrates CDC25A may or may not recognise in the cellular environment, thereby providing information beyond just the complex studied structurally.

The manuscript is generally well-written, though occasionally a bit heavy on description of structural detail. The structural work has been conducted carefully, and the effort of the authors to obtain cryo-EM data with full angular coverage and an isotropically resolved map should be commended.

I have no serious concerns regarding the validity of the findings, which are well-supported by the structural data, but presentation and description of some of the technical aspects of the cryo-EM work need more elaboration. I have summarised these aspects below.

The manuscript by Rowland et al. is a strong candidate for publication in Nature Communications.

Specific comments:

- The resolution value of 2.91 Å used throughout the manuscript exaggerates the accuracy of resolution determination. This should be revised to 2.9 Å and all other resolution values in the manuscript should be rounded to tenths of Å as well.

- It appears that the authors have chosen a relatively simple cryoSPARC-only strategy for refinement of their cryo-EM map. Looking at the existing literature, it would appear that the use of RELION to enable better 3D classification and particle polishing could potentially provide better results for the type of complex the authors have investigated. Exploring this avenue may be worthwhile, particularly considering the claim of technical excellence made near the bottom of page 4. However, I want to emphasize that the density as-is already supports the authors' conclusions, and that this suggestion is not meant to be a requirement for publication.

- Page 9, recognition of pTyr15 - while the cryo-EM density of the putative phosphate is quite strong, the density in this area (G-rich loop), which includes Tyr15, is fragmented. While I agree that the proposed model is most likely correct, the authors should disclose this fact in the text at this point, even if only with a brief statement (they do discuss this later, on page 10).

- Methods: i) Exposure time of 6.5 sec at the given magnification implies very low dose rate - please confirm this is correct. ii) "Micron" = micrometer (probably best to use the correct abbreviation using lower-case greek letter mu; further down uL using the letter u is used, which should also be corrected).

- The energy filter type is currently given as "GIF energy filter" (GIF = "Gatan Imaging Filter") in Supplementary Figure 1. As the authors note in the main text, the detector is a K3 and the energy filter type is BioQuantum. The sentence near the bottom of page 6 and on page 25 ("BioQuantum K3 detector and GIF energy filter") may need adjusting to clarify what is what as well.

- In Supplementary Table 1, the authors state a "model resolution range" of 2.7-3.4 Å. The most commonly used validation tools will provide a model vs. map FSC curve (which should be shown in the paper but currently is not) that provides a single number. So, how was the "model resolution range" determined? Did the authors run a local model vs. map FSC calculation? If not, this should be changed to a single value indicating just the resolution at which the model vs. map FSC passes below 0.5.

Minor (including typos):

Figure 1f, legend - "the local resolution ranges 2.4-3.0 Å" sounds odd.

Figure 1g - text claims residues 335-524 were modelled, the figure suggests 523 is the C-terminal residue (text on page 9 appears to reaffirm 523 as the end of the model, but a Leu524 is actually contained in the model). Please clarify.

Page 20 - "This residue is conserved in CDK6 but not in CDK4, in which Gln131 is mutated to glutamate." This residue is not truly "mutated" - it is just a glutamate.

Reviewer #3 (Remarks to the Author):

This manuscript reports the first structure of a CDC25 family phosphatase (CDC25) in complex with a cyclin dependent kinase (CDK2-CYCA2). These proteins are central regulators of cell division, and how they interact has been a critical knowledge gap in the cell cycle field with high relevance to cancer therapeutic development, as CDKs have become highly studied targets. The authors use a clever experimental strategy to trap the ternary complex and determine the structure using cryo-electron microscopy. The data analysis and final structural model are of high technical quality, and the structural conclusions, which are all quite well explained, are fairly drawn. Sufficient details are given in the methods section to reproduce the results. The manuscript warrants publication with very minor revisions in Nature Communications. However, the biological insights the structure provides could potentially be expanded with more discussion and perhaps a few more straightforward experiments. Below are some questions and comments.

1) The observation that the binding sites of CDC25 and CKS1 overlap is quite interesting, and this relationship has not been addressed very well in the literature. It would be exciting if the authors could demonstrate that CKS1 inhibits CDC25 binding to and/or dephosphorylation of CDK2. At least, the authors could discuss the biological implications of this competition. For example, do the authors envision that only CDK1/2 complexes lacking CKS1/2 get dephosphorylated in the cell? Does this requirement suggest something about the order of CDK activation by CDC25 and CKS1?

2) It is also interesting that CDC25 contacts both CDK2 and CYCA. It would be interesting to validate that the affinity for CDK2 alone is weaker and discuss possible implications for order of activation in the cell.

3) The rationale for and conclusion from the HD-MX experiment are a little difficult to follow from the main text of the results section. Perhaps the authors could more explicitly state why they performed this analysis (e.g. it was to validate the interface elements observed in the EM structure) and what is the main takeaway. From the methods, it appears that the CDC25A WT and not active site mutant was used. Authors should add this point to the results and discuss whether perhaps this was also part of the motivation of the solution study. Also, the suggestion that a lack of protection of the CDC25A C-helix was observed because it is dynamic is reasonable, but then one might expect this to be reflected in the resolution/quality of the EM map in that region. Is this the case? It is a bit difficult to see the location of the C-terminal helix in the local resolution map.

4) Similarly, it should be made clear whether the C431S mutant was used in the HTRF experiments or not. If not, it would be interesting, though not essential, to know how the WT vs. mutant affinity compares. Also, authors could report the K_d determined in the Supplementary Figure 6 experiment.

5) The argument about the relevance of the *S. pombe* *cdc2* mutations in the GDSEID motif was not entirely clear. Do the authors mean that because the yeast arrest at G2/M that these mutations are only relevant to CDC25 and not CKS1 or KAP function? The authors could remind readers what the different phenotypes are for the different genetic knockouts. Biochemical evidence supporting that those mutations specifically inhibit CDC25 and not the other proteins would be important to make that claim. Alternatively, perhaps the comparison is not so vital to the thrust of the paper, and the mechanistic explanation for the mutations could be moved instead to follow the TR-FRET data on CDC25 binding. In other words, the mutation data could be used simply to compliment the structural observation about where CDC25 binds, rather than to make a subtle (and not well substantiated) point about differences in CDC25 and CKS1 or KAP binding.

6) There is some evidence in the literature that Cdk4 is phosphorylated in Y19 and that this phosphate is removed for G1/S progression, e.g. PMID: 7630405. There is less evidence that CDC25 is the phosphatase that removes the inhibitory phosphate in cells. The authors could more thoroughly discuss this literature, and if possible, test the prediction from the structure that CDK4/6 indeed binds with less affinity and/or dephosphorylate with less efficacy than CDK2.

7) Page 18, first full paragraph, first sentence beginning "Whilst": Do the authors mean the last word of that sentence to be "CKS1" and not "KAP"?

8) It would be helpful if the authors chose to show 2D classes with comparable orientations across the different conditions in Fig. 1a-d. For example, placing the similar orientations by column would allow the reader to gain a better appreciation of the differences in quality. Additional orientations could be shown with CHAPS to convey that more orientations were imaged in the presence of detergent. Also, do the authors intend to show the same 2D class panel in Fig. 1 and Sup. Fig 2?

9) If possible, keeping the same orientation of the complex in Fig. 1 and 2 would be helpful.

We would like to thank all three reviewers for taking the time to review the MS and for their helpful suggestions and comments.

Reviewer #1 (Remarks to the Author):

In this manuscript Rowland et al. present a cryo-EM structure of the CDK2-cyclin A-CDC25A complex. Overall, this manuscript is well written, the cryo-EM work is of high-quality and the scientific finding is of high interest.

CDC25 phosphatases regulate the cell cycle by removing inhibitory phosphorylation on the G-loop of CDK-cyclin complexes, thereby activating the CDK-cyclin complexes, which are master regulators of the cell cycle. Although the structure of CDC25 proteins on its own was already known, in this work the authors managed to obtain a snapshot of a CDK-cyclin-CDC25 complex thereby elucidating its stoichiometry, mechanisms of assembly and activation. These findings are important for the mechanistic understanding of the cell cycle, which is a process pivotal for life and deregulated in diseases. For all of these reasons I would recommend publication of this manuscript in Nature Communications. I have only a few minor comments with the hope to improve the quality of this article even further:

1) Page 4 “one of the smallest asymmetric particles to have been solved by cryo-EM”; cryo-EM structures are determined and not solved as in X-ray crystallography, therefore “solved” need replacing with “determined”.

We thank the reviewer for noting this point and have changed the text (underlined) on p4 to read:

“Together these proteins constitute an ~ 86 kDa complex, making it among the smallest of asymmetric particles to have been determined by cryo-EM to a resolution better than 3 Å”

2) Top of page 5, comparisons between the authors’ cryo-EM structure and other CDKs cyclins: I appreciate the sequence conservation analysis performed by the authors, however I am wondering if AlphaFold multimer could be used to support these predictions further. The AlphaFold predictions could also help enriching the discussion/conclusion section (last paragraph of the main text, pages 18-22).

We thank the reviewer for this suggestion. We have now used AlphaFold Multimer to explore potential CDC25A-CDK4-cyclin D1/D3 complex structures. The rationale for this study and the added text is provided in full in response to Reviewer 3, point 6 who also asked this question. Briefly, we assessed CDK4-cyclin D1/D3 complexes as there is evidence in the literature that CDK4 is phosphorylated on Tyr17 (the residue equivalent to CDK2 Tyr15) in response to uv irradiation (see PMID: 7630405, ref 73 in the MS). We have incorporated the AlphaFold multimer results as Supplementary Fig. 8 and the accompanying text is included in full below in response to Reviewer 3, point 6.

3) Pages 6 and 7. Most of the paragraph titled “Sample preparation and preliminary data collection” could be moved in the Method section as it contains very technical details (e.g. ModelAngelo was used for...), which are interesting for the cryo-EM reader and not for the general reader. This excess of technical details in the main text may obscure the biological findings coming from the structure.

We agree with the Reviewer 1 that some of the methods text at the start of the Results section could be moved into the Methods section. To accommodate this comment and that on that same topic made by Reviewer 2, we have removed the detailed description of how the complex was assembled and merged the first two subsections into one entitled “Data collection and structure determination”. Assembly of the complex is rather straightforward and is described in detail in the Supplementary Information. This change we think then makes the start of the Results and Discussion section rather succinct. We would like to retain the text that describes the use of the ModelAngelo software as we believe this software should be highlighted. The revised text (on p6) now reads:

Results and Discussion

Data collection and structure determination

The CDK2-cyclin A-CDC25A complex was phosphorylated on CDK2 Tyr15 and Thr160, and the CDC25A was catalytically inactivated by mutating Cys431 to serine to generate a stable ternary complex. (See Supplementary Methods and Supplementary Fig. 1 for further details). Preliminary data collection on a 200 kV Glacios.....

4) Page 9, comparing with PDB 1C25A, RMSD value should be indicated. In the following sentence, PTP C(X)₅R loop could be depicted in some figure and Cys431Ser and pTyr15 should be referenced to Figure 3. In general referencing and labelling in this paragraph should be improved.

(i) To include the RMSD value the text now reads:

The CDC25A catalytic domain structure is broadly consistent with the monomeric crystal form (PDB 1C25A³⁷) comprising an α/β -domain with a central 5-stranded parallel β -sheet (of strand order 15423) enclosed by 5 α -helices (RMSD of aligned residues (335-524) = 0.9 Å), (Fig. 2c).

(ii) The PTP C(X)₅R loop is drawn in Figure 3a but was not explicitly coloured. To address this comment, we have coloured the sequence yellow in Figure 3a and edited the legend.

(iii) We apologise for the omissions in the Figure 3 labelling. These are now corrected: Cys431 (a serine in our structure) and pTyr15 are drawn and labelled in Figure 3 panels (a) and (c). To highlight these residues, we now mention them in a revised Figure 3 legend. Text now reads:

(a) Overlay of CDK2 (purple) with pT160CDK2-cyclin A crystal structure (PDB 1JST, grey), highlighting a change in G-loop conformation. The CDC25A CX5R loop is highlighted in yellow.

(c) Ribbon and density diagram (0.06 threshold) for the binding of CDK2 pTyr15 in the CDC25A active site. pTyr15 and the mutated catalytic cysteine residue (Cys431Ser) are labelled in panels (a) and (c).

(iv) We have also reviewed the referencing and labelling in this paragraph and made the following changes to address the reviewer’s concerns:

Identified the CDC25A β -sheet and α -helix that bookend the catalytic loop. Text now reads (p9):

....the conserved PTP C(X)5R loop between a central β -strand (β 4) and α -helix (α 4), creating....

Labelled the CDC25A secondary structural elements in Fig. 2c and labelled the locations of the secondary structural elements in the CDK, cyclin and CDC25 sequence alignments in Supplementary Fig. 7.

Identified the cyclin A C-CBF helix that binds to the CDC25A C-terminal tail. Text now reads (p9):

.....protein-protein interface of $\sim 170 \text{ \AA}^2$ mediated by the loop linking α 3' to α 4' in the C-CBF.

The CBF helices α 3' and α 4' are now labelled in Fig. 5b.

5) Page 10, beginning of paragraph CDK2-CDC25A interactions... Similar issues with referencing and labelling as in point 4. Can the G-loop be depicted better perhaps in Figure 3? GDSEID-alphaGhelix motif, Figure 3 should be referenced.

We thank the reviewer for these comments and for pointing out that we have not introduced the GDSEID motif. The CDK2 G-loop is depicted in Fig. 3a-c. To address these comments, we have:

(i) Labelled the G-loop in Fig. 3a.

(ii) Changed the text at the end of the Introduction (on p5) to read:

"...indicate the GDSEID motif (single letter amino acid code, residues 205-210 in CDK2), which is conserved in CDKs 1-3 and is located in the CDK C-terminal lobe, is a mutual binding site for these proteins that also imparts selectivity, highlighting its importance as a CDK regulatory site."

(iii) We have moved the text that introduces the GDSEID motif from the start of the section entitled "The CDK2 GDSEID sequence mediates CDK2-protein interactions" up to the last paragraph in the section entitled "CDC25A-CDK2 interactions: CDC25A substrate recognition". Introducing the text here allows some shortening to avoid duplication. The text (on p13) now reads:

"Within the CDK C-terminal lobe, the CDK family has a loop linking the α F and α G helices that contains the GDSEID sequence that is conserved in CDKs 1-3. The importance of this motif to CDK1 function was first recognised through the isolation of *Schizosaccharomyces pombe cdc2* cell cycle mutants (compiled in Endicott et al.⁶¹). The α G helix and preceding GDSEID sequence protrude over the CDC25A helix that lies C-terminal to the mutated catalytic Cys431Ser residue. This....."

(iv) To accommodate the text move above, the start of the section entitled "The CDK2 GDSEID sequence mediates CDK2-protein interactions" (on p16) now reads:

With the determination of the structure of CDK2-cyclin A-CDC25A there are now structures for three protein complexes that exploit the CDK2-C-terminal lobe for complex formation.

The structures of CDK2 bound to cyclin-dependent kinase subunit (CKS)1 (PDB 1BUH⁵⁰) and kinase associated phosphatase (KAP) (PDB 1FQ1⁴⁹) were the first to illustrate the importance of the GDSEID motif to CDK1/2 regulation. ~~This motif is not conserved in other CDKs (Fig. 6a) and is extended by 9 residues in CDK8 (residues 240-248)."~~

The last sentence is deleted as the CDK8 comparison is not relevant, and a more detailed comparison with CDK4 and CDK6 is included later in the MS.

6) Page 15, bottom paragraph, again similar issue, Thr507 should be referenced to some Figure... maybe depict in Figure 5c? The result of mutation Thr to Glu should be referenced to something?

We introduced a glutamate at position 507 as a mimetic for a phosphorylated threonine. PhosphoThr507 is proposed to be one of two phosphorylated residues that form a bidentate interaction with 14-3-3 proteins.

To address this comment, we have included:

- (i) the reference for 14-3-3 interaction with CDC25A
- (ii) the Thr507 sidechain in Figure 5c;
- (iii) Included a reference to Supplementary Fig. 6 (HTRF analysis)

This last paragraph (now on p16) reads:

The C-terminal sequence has previously been shown to mediate the interaction of CDC25A with 14-3-3 proteins. 14-3-3 proteins recognize CDC25A phosphorylated on Thr507 as one of two phosphoamino acids required for their bidentate interaction with ligands^{40,42}. In our cryo-EM structure, Thr507 points out into solution, with no apparent CDK2-cyclin A interactions (Fig. 5c). As expected, analysis by HTRF revealed mutation of Thr507 to a glutamate (as a phosphothreonine mimetic) had little effect on the affinity of CDC25A for CDK2-cyclin A (Supplementary Fig. 6d). To be a 14-3-3 ligand, the CDC25A tail is most likely flexible in solution to promote first recognition by the CHK1 catalytic site and then subsequent binding to the extended 14-3-3 phospho-peptide recognition cleft. This function of CDC25A further supports our hypothesis that the C-terminal region of CDC25A must be structurally dynamic but forms helical secondary structure on binding to the CDK2-cyclin A substrate.

Reviewer #2 (Remarks to the Author):

Comments on Rowland et al.:

The eukaryotic cell cycle is controlled by an intricate regulatory system that ensures faithful execution of critical cellular processes in a coordinated fashion. Cyclin-dependent kinases (CDKs) are at the core of these regulatory circuits. Therefore, they are themselves subject to multiple layers of regulation, including by association with protein partners (such as cyclins) and by post-translational modifications, most notably phosphorylation and dephosphorylation of key residues. In their manuscript, Rowland et al. describe the cryo-EM analysis of the CDK2-cyclin A complex bound to the phosphatase CDC25A, which dephosphorylates and thereby de-represses CDK2 and other cyclin-dependent kinases. The work described by Rowland et al. reveals the mechanism by which CDC25A recognises CDK2-cyclin A. Sequence comparisons suggest which additional substrates CDC25A may or may not

recognise in the cellular environment, thereby providing information beyond just the complex studied structurally.

The manuscript is generally well-written, though occasionally a bit heavy on description of structural detail. The structural work has been conducted carefully, and the effort of the authors to obtain cryo-EM data with full angular coverage and an isotropically resolved map should be commended.

I have no serious concerns regarding the validity of the findings, which are well-supported by the structural data, but presentation and description of some of the technical aspects of the cryo-EM work need more elaboration. I have summarised these aspects below.

The manuscript by Rowland et al. is a strong candidate for publication in Nature Communications.

Specific comments:

- The resolution value of 2.91 Å used throughout the manuscript exaggerates the accuracy of resolution determination. This should be revised to 2.9 Å and all other resolution values in the manuscript should be rounded to tenths of Å as well.

We appreciate the reviewer's point and now state all resolutions to 1DP. Since submission we have refined the structure to 2.7 Å and have changed the resolution value throughout. The local resolution analysis has been repeated with the new map and Figures 1, 3, 4 and 5 and Supplementary Figs 3 and 4 (and accompanying Figure Legends) have been updated to display density of the new map with . Table S1 has been updated to present the revised refinement and validation statistics. On p7, the text now reads:

The local resolution within the complex varies from 2.2 Å to 2.8 Å (Fig. 1f), with the highest resolution within the CDK2-cyclin A core,

- It appears that the authors have chosen a relatively simple cryoSPARC-only strategy for refinement of their cryo-EM map. Looking at the existing literature, it would appear that the use of RELION to enable better 3D classification and particle polishing could potentially provide better results for the type of complex the authors have investigated. Exploring this avenue may be worthwhile, particularly considering the claim of technical excellence made near the bottom of page 4. However, I want to emphasize that the density as-is already supports the authors' conclusions, and that this suggestion is not meant to be a requirement for publication.

We would like to thank the reviewer for this suggestion. We agree that Relion has a strong reputation for 3D classification and post-processing options. However, from our experience, cryoSPARC has proved more powerful for particle picking and 2D classification, with recent improvements in 3D classification. We did perform preliminary data processing in Relion 4 but this did not indicate any significant improvements in the 3D reconstruction and 2D classification proved more tricky. We also encountered issues in the data interconversion between cryoSPARC and Relion, leading us to choose a cryoSPARC only approach that we feel has yielded suitable results. Regarding the suggestion of particle polishing, cryoSPARC has recently released its reference-based per-particle motion correction job and we have employed this to improve the resolution of our cryo-EM map from 2.9 Å to 2.7 Å. The model has been docked and refined in the new map (giving an RMSD of 0.06 between the old and

new models) and a model-map FSC has been generated (Supplementary Fig. 4e). We have deposited the new map and model in place of our original submission.

- Page 9, recognition of pTyr15 - while the cryo-EM density of the putative phosphate is quite strong, the density in this area (G-rich loop), which includes Tyr15, is fragmented. While I agree that the proposed model is most likely correct, the authors should disclose this fact in the text at this point, even if only with a brief statement (they do discuss this later, on page 10).

We agree with the reviewer that this is an important point to make. However, this section is entitled "Overview of the CDK2-cyclin A-CDC25A complex" and we think that a more detailed description of the nature of the engagement of the CDK2 G-loop with the CDC25A catalytic site sits more comfortably in the section entitled "CDC25A-CDK2 interactions: CDC25A substrate recognition".

We hope that the following text change (at the bottom of p10) addresses this comment:

"Within the CDC25A active site, clear density for CDK2 pTyr15 was observed (Fig. 3a-c). Although density for the surrounding CDK2 residues was fragmented so that sidechain locations were more ambiguous, the backbone structure for all residues of the G-loop could be modelled."

- Methods: i) Exposure time of 6.5 sec at the given magnification implies very low dose rate - please confirm this is correct.

This exposure time is correct, please see screenshot of data collection record below:

```
User: Rhianna Rowland
Sample: XXXXXXXXX
Date: 2022-11-18

Microscope Settings
Voltage (kV): 200
GL (arb): 4

Data Acquisition Settings
Total dose (e-/A2): 50
Exposure time (seconds): 6.58
Number of EER frames: 1582
Fluence over vacuum (e-/pix/s): 2.75
Fluence over vacuum (e-/A2/s): 8.31
Probe Mode: NanoProbe
Mag: 240k
Spot Size: 4
Intensity: 0.442
C2 aperture (microns): 50
Calibrated pixel size (Angstrom): 0.574
Illum. area (microns): 1.8

Acquisition Mode (AFIS/normal): AFIS
```

ii) "Micron" = micrometer (probably best to use the correct abbreviation using lower-case greek letter mu; further down uL using the letter u is used, which should also be corrected).

We apologise for this typographical error. All instances of u instead of μ have been corrected.

- The energy filter type is currently given as "GIF energy filter" (GIF = "Gatan Imaging Filter") in Supplementary Fig. 1. As the authors note in the main text, the detector is a K3 and the

energy filter type is BioQuantum. The sentence near the bottom of page 6 and on page 25 ("BioQuantum K3 detector and GIF energy filter") may need adjusting to clarify what is what as well.

We thank the reviewer for pointing out this error. This point is now corrected in the main text (p6) to read:

"High-resolution data collection on grids prepared with 0.5 and 1.0 X CMC CHAPS was performed on a 300 kV FEI Titan Krios equipped with a BioQuantum K3 detector and ~~GF~~ energy filter (Supplementary Table 1)."

And in the Methods (p26) to read:

"Cryo-EM images were acquired on a 300 kV FEI Titan Krios equipped with a BioQuantum K3 detector and imaging filter, housed at the Electron Bio-Imaging Centre (Krios 1, eBIC, Diamond Light Source, UK)."

- In Supplementary Table 1, the authors state a "model resolution range" of 2.7-3.4 Å. The most commonly used validation tools will provide a model vs. map FSC curve (which should be shown in the paper but currently is not) that provides a single number. So, how was the "model resolution range" determined? Did the authors run a local model vs. map FSC calculation? If not, this should be changed to a single value indicating just the resolution at which the model vs. map FSC passes below 0.5.

We thank the reviewer for this comment. To address it we have included a map-model FSC as Supplementary Fig. 4e. The accompanying legend reads:

"(e) A model-map FSC was generated using the comprehensive validation job in Phenix, showing the Fourier shell coefficient curve based on the model map with and without masking (orange and blue lines respectively). The intersections of the curves with FSC=0.5 are shown."

On p28, the last paragraph in the section entitled "Data processing, and structure determination" now reads:

"Particles from the best 3D heterogenous class (670,852 particles) were re-extracted without Fourier cropping to give an effective pixel size of 0.825 Å and refined by homogeneous refinement with a 10 Å maximum alignment resolution. The 3D reconstruction was further refined by non-uniform refinement⁷⁷ to yield a 2.9 Å reconstruction of the trimeric complex. This reconstruction was used as a reference to perform per-particle motion correction⁷⁸ in cryoSPARC, after which non-uniform refinement was repeated with the polished particle set to yield the final 2.7 Å resolution EM map. The local resolution variation was estimated via the local resolution estimation job at 0.143 FSC threshold."

We have added the following text to the Supplementary Fig. 3 legend:

"This reconstruction was used as a reference to perform per-particle motion correction of the final particle set. Non-uniform refinement of the polished particle set yielded a 2.7 Å cryo-EM map of the trimeric complex (FSC 0.143). "

Minor (including typos):

- Figure 1f, legend - "the local resolution ranges 2.4-3.0 Å" sounds odd.

To address this comment the text now reads:

"The local resolution within the complex varies from 2.2 to 2.8 Å"

- Figure 1g - text claims residues 335-524 were modelled, the figure suggests 523 is the C-terminal residue (text on page 9 appears to reaffirm 523 as the end of the model, but a Leu524 is actually contained in the model). Please clarify.

We apologise for this error. L524 is the sequence C-terminal residue and has associated density in the map. We have corrected the terminal residue to 524 in text.

Page 20 - "This residue is conserved in CDK6 but not in CDK4, in which Gln131 is mutated to glutamate." This residue is not truly "mutated" - it is just a glutamate.

Thank you. The text (now on p21) is changed to:

"This residue is conserved in CDK6 but not in CDK4, in which Gln131 is a glutamate."

Reviewer #3 (Remarks to the Author):

This manuscript reports the first structure of a CDC25 family phosphatase (CDC25) in complex with a cyclin dependent kinase (CDK2-CYCA2). These proteins are central regulators of cell division, and how they interact has been a critical knowledge gap in the cell cycle field with high relevance to cancer therapeutic development, as CDKs have become highly studied targets. The authors use a clever experimental strategy to trap the ternary complex and determine the structure using cryo-electron microscopy. The data analysis and final structural model are of high technical quality, and the structural conclusions, which are all quite well explained, are fairly drawn. Sufficient details are given in the methods section to reproduce the results. The manuscript warrants publication with very minor revisions in Nature Communications. However, the biological insights the structure provides could potentially be expanded with more discussion and perhaps a few more straightforward experiments. Below are some questions and comments.

1) The observation that the binding sites of CDC25 and CKS1 overlap is quite interesting, and this relationship has not been addressed very well in the literature. It would be exciting if the authors could demonstrate that CKS1 inhibits CDC25 binding to and/or dephosphorylation of CDK2. At least, the authors could discuss the biological implications of this competition. For example, do the authors envision that only CDK1/2 complexes lacking CKS1/2 get dephosphorylated in the cell? Does this requirement suggest something about the order of CDK activation by CDC25 and CKS1?

We thank the reviewer for these insightful comments. The binding of CKS1 and CDC25A to CDK2-cyclin A is mutually exclusive. When we first attempted to determine the structure by cryo-EM (repeated crystallization trials having failed), we tested whether we could increase the size of the complex by making a quaternary CDK2-cyclin A-CKS1-CDC25A complex. However, mixing CDK2-cyclin A-CDC25A with excess CKS1 or CDK2-cyclin A-CKS1 with excess CDC25A and then analyzing by size-exclusion chromatography always generated a

redistribution of the CDC25A and CKS1 between CDK2-cyclin A bound and unbound populations reflecting their affinities and the SEC experimental parameters. Our results would suggest that CDK1/2-cyclin complexes associated with CKS1 would not be substrates for CDC25. The interaction between CDK1/2 and CKS1 is a protein-protein interaction and that between CDK1/2 and CDC25 a catalytic engagement (that we stabilise by employing a substrate-trapping mutant strategy).

To answer this comment and this reviewer's comment 2 (below), we carried out surface plasmon resonance (SPR) to assess the binding of CDC25A(C431S) to monomeric pT160CDK2, pY15pT160CDK2 and pT160CDK2-cyclin A. To address the observation that the CKS1 and CDC25A binding sites overlap on CDK2, we included a comparison of the binding of CDC25A(C431S) to pT160CDK2-cyclin A and to pT160CDK2-cyclin A-CKS1. Using SPR, the binding of pT160CDK2-cyclin A and pT160CDK2-cyclin A-CKS1 to immobilized GSTCDC25A(C431S) is indistinguishable. In hindsight though SPR is appropriate to assess the interaction of CDC25A with monomeric CDK2 and CDK2-cyclin A complexes, given the flow rate and association time used in these experiments we would expect significant dissociation of CKS1 from pT160CDK2-cyclin A such that the species binding to CDC25A(C431S) is the same in each case. We include the SPR sensorgram for the binding of pT160CDK2-cyclin A-CKS1 to immobilized GST-CDC25A here for review (see below). The apparent K_d is $16 \pm 1.9 \mu\text{M}$, ie similar to that determined for the binding of pT160CDK2-cyclin A to CDC25A (K_d , $15 \pm 1.2 \mu\text{M}$). The composition of the various CDK2 species used in our SPR analysis is shown in the RHS panel below. The sensorgram for the binding of pT160CDK2-cyclin A to CDC25A is included in Supplementary Material as Supplementary Fig. 6c. Our SEC and SPR analyses are compatible with our conclusion drawn from a comparison of the two ternary complexes that CKS1 and CDC25A binding to CDK2-cyclin A is mutually exclusive. Quantitative analysis of their competitive binding will require further experiments that we will pursue, but that we would request are beyond the scope of the current manuscript.

To address the biological implications of this competition, we note that CDC25 is a CDK1 substrate and CKS1 binding to CDK1 has been shown to promote CDC25 multisite phosphorylation. It has been proposed that this interaction would be promoted by the CDC25 N-terminal RXL motif binding to the cyclin A/B recruitment site.

Whether competition between CKS1 and CDC25 for binding to CDK1 or CDK2 is a mechanism to regulate CDK1/2 activity either in response to checkpoint signaling or to

order cell cycle progression is a question that would require additional experimentation beyond the scope of this paper. However, the structure does provide a starting point to design experiments to address this question, and as suggested by the reviewer we speculate on the possible functional importance of CKS1 and CDC25 CDK binding being mutually exclusive. We have added the following text on p18/19:

“CKS1 is important for CDK1 and CDK2 substrate recognition. When bound to CDK2 within a pentameric CDK2-cyclin A-CKS1-SKP1-SKP2 complex, CKS1 recognizes the phosphoThr187 of p27KIP1 which is then recruited to the SCF^{SKP2} E3 ubiquitin ligase complex^{64–66}. The binding site for CKS1 on CDK2 is also conserved in CDK1^{67,68}, and CKS1 promotes CDK1-cyclin A/B multi-site phosphorylation of substrates that include CDC25, WEE1 and MYT1^{69,70}. From the structural comparison, it can be hypothesized that CKS1 binding to CDK1 would impact the reciprocal regulation of CDK1 and CDC25. Association of CDK1 with CKS1 would preclude subsequent regulation of CDK1-cyclin A/B by active site phosphorylation by blocking CDC25 binding (and possibly WEE1/MYT1 binding) while potentially promoting the multisite phosphorylation of CDC25 within its N-terminal sequence. A comparison of the CDK1/2-CKS1 and CDK1/2-CDC25A interfaces provides an opportunity to identify CDK mutations that might distinguish CKS1 and CDC25A binding to explore their mutual regulation of CDK1 and CDK2 activities.

In summary, it is evident from a comparison of CDK2 bound to KAP, CKS1 and CDC25A that...”

2) It is also interesting that CDC25 contacts both CDK2 and CYCA. It would be interesting to validate that the affinity for CDK2 alone is weaker and discuss possible implications for order of activation in the cell.

The accepted model is that the substrate for WEE1 kinase is pT160CDK2-cyclin A and the substrate for CDC25 is pY15pT160CDK2-cyclin A, though the CDC25A substrate trapping mutant (C431S) has appreciable affinity for pT160CDK2-cyclin A (as we show in Supplementary Fig. 6). We attempted to bind CDC25A to unphosphorylated CDK2-cyclin A (result not shown) and detected no interaction, confirming the importance of the conformation of the CDK2 activation segment for CDC25A substrate recognition.

The structure of monomeric pT160CDK2 (PDB: 1B39) would also be predicted to be incompatible with CDC25A binding given the activation segment conformation. In this structure the CDK2 activation segment is folded to partially occlude the G-loop, however the quality of the electron density map in this region suggests that this sequence is flexible and may adopt several different mobile states. pT160CDK2 residues 26-43 that precede the C-helix are disordered and could not be built.

To address the reviewer’s comment, we have used SPR (coupling GSTCDC25A(C431S) to the chip) to compare the affinity of CDC25A(C431S) for pY15pT160CDK2, pT160CDK2 and pT160CDK2-cyclin A. The results are included in the manuscript in a modified Supplementary Fig. 6. We detect binding between CDC25A(C431S) and pT160CDK2-cyclin A that yields an apparent affinity of 15 μ M (compared to an affinity of $0.11 \pm 0.02 \mu$ M for the affinity of CDC25A(C431S) to pT160CDK2-cyclin A determined by HTRF (now shown in Supplementary Fig. 6d). There was no detectable binding between CDC25A(C431S) and either mono or doubly phosphorylated monomeric CDK2 (shown in Supplementary Fig.

6a,b). We conclude that a significant role of cyclin A (together with CDK2 activation segment phosphorylation) is to remodel CDK2 into a conformation around the active site that makes CDK2 a substrate for CDC25A. We have added text to address point 2) starting on p15 as follows:

“To explore the requirement for cyclin A for complex formation, we employed surface plasmon resonance to compare the affinity of CDC25A(C431S) for monomeric pY15pT160CDK2, pT160CDK2 and for pY15pT160CDK2-cyclin A. Despite making few interactions with cyclin A, the affinity of CDC25A(C431S) for pT160CDK2-cyclin A ($K_d = 15 \pm 1.2 \mu\text{M}$) is significantly greater than that for monomeric pY15pT160CDK2 and pT160CDK2 (for which K_d values could not be reliably determined, Supplementary Fig. 6a-c). The structure of monomeric pT160CDK2 (PDB **1B39**⁶²) is predicted to be incompatible with CDC25A binding because the CDK2 activation loop is folded to partially occlude the G-loop. The quality of the electron density map in this region of the monomeric pT160CDK2 crystal structure suggests this sequence is flexible and may adopt several different mobile states, whilst CDK2 residues 36-43 that precede the C-helix are also disordered and could not be modelled. We conclude that cyclin A has a significant role in facilitating the formation of the CDK2-cyclin A-CDC25A complex by enforcing changes in CDK2 conformation around the active site to make CDK2 a CDC25A substrate.”

We have also added (i) sections in the Methods to describe production of pT160CDK2 for the SPR experiments and SPR methodology and (ii) Natalie J. Tatum to the author list to reflect her contribution to the completion of these experiments.

To reflect the above edit, we have changed the title of the next sub-section from “The CDC25A C-terminal helix mediates complex formation” to “The CDC25A C-terminal helix is required for complex formation”.

3) The rationale for and conclusion from the HD-MX experiment are a little difficult to follow from the main text of the results section. Perhaps the authors could more explicitly state why they performed this analysis (e.g. it was to validate the interface elements observed in the EM structure) and what is the main takeaway. From the methods, it appears that the CDC25A WT and not active site mutant was used. Authors should add this point to the results and discuss whether perhaps this was also part of the motivation of the solution study. Also, the suggestion that a lack of protection of the CDC25A C-helix was observed because it is dynamic is reasonable, but then one might expect this to be reflected in the resolution/quality of the EM map in that region. Is this the case? It is a bit difficult to see the location of the C-terminal helix in the local resolution map.

We performed the HDX-MS analysis before we determined the structure. The study was to better understand how the CDC25A tail bound to cyclin A and to map the CDC25A binding sites on CDK2 and cyclin A to identify further residues to mutate to probe function. Our early work showed that the CDC25A tail was required for complex formation as the Δ Ctail mutant does not bind to CDK2-cyclin A. Given the lack of secondary structure prediction for this sequence our working model was that it might adopt an extended conformation upon cyclin A binding resembling the binding mode previously observed with p27KIP1 and SKP2. The HDX-MS was carried out to assess this model and to also map the CDC25A-CDK2

interaction. We did not expect the CDC25A tail to adopt a helical conformation on binding. We appreciate that this timeline does not come through in the current text, and have changed the MS (on p9) to now read:

“Prior to our determination of the structure of the CDK2-cyclin A-CDC25A complex we analyzed it by hydrogen-deuterium exchange mass spectrometry (HDX-MS) comparing monomeric CDC25A against the trimeric complex (Supplementary Fig. 5). In agreement with the structure, this analysis highlighted a region of CDC25A that is significantly.....

The last paragraph of this section we would like to keep as the HDX-MS data does suggest that “the C-terminal helix is not substantially solvent-protected in the bound state”, a statement supported by the structure. We also suggest that the experiment provides evidence in support of the hypothesis that the C-terminal sequence is structurally dynamic in solution. That the C-terminal sequence is structurally dynamic is evidenced by the crystal structure of monomeric CDC25A whereas as we illustrate in Figure 2c, the sequence adopts an extended conformation to residue 495 and then is flexible.

This ability to adopt an extended conformation would also be a requirement for binding to 14-3-3 proteins. A point we now bring out more clearly in the section entitled “The CDC25A C-terminal helix is required for complex formation”:

Despite making few interactions with CDC25A, we confirmed the requirement for the CDC25A C-terminal tail for binding to CDK2-cyclin A by assaying a set of CDC25A C-terminal tail mutants through homogenous time-resolved fluorescence (HTRF) (Supplementary Fig. 6d). By HTRF the affinity of CDC25A(C431S) for pT160CDK2-cyclin A was $0.11 \pm 0.02 \mu\text{M}$.

The C-terminal sequence has previously been shown to mediate the interaction of CDC25A with 14-3-3 proteins. 14-3-3 proteins recognize CDC25A phosphorylated on Thr507 as one of two phosphoamino acids required for their bidentate interaction with ligands^{40,42}. In our cryo-EM structure, Thr507 points out into solution, with no apparent CDK2-cyclin A interactions (Fig. 5c). As expected, analysis by HTRF revealed mutation of Thr507 to a glutamate (as a phosphothreonine mimetic) had little effect on the affinity of CDC25A for CDK2-cyclin A (Supplementary Fig. 6). To be a 14-3-3 ligand, the CDC25A tail is most likely flexible in solution to promote first recognition by the CHK1 catalytic site and then subsequent binding to the extended 14-3-3 phospho-peptide recognition cleft. This function of CDC25A further supports our hypothesis that the C-terminal region of CDC25A must be structurally dynamic, but forms helical secondary structure on binding to the CDK2-cyclin A substrate.

4) Similarly, it should be made clear whether the C431S mutant was used in the HTRF experiments or not. If not, it would be interesting, though not essential, to know how the WT vs. mutant affinity compares. Also, authors could report the K_d determined in the Supplementary Figure 6 experiment.

We apologise for this omission. The C431S mutant was used in the HTRF and HDX-MS experiments. The text has now been corrected. We now report the K_d determined by HTRF ($0.11 \pm 0.02 \mu\text{M}$) in both the text and the Supplementary Fig. 6 figure legend.

5) *The argument about the relevance of the S. pombe cdc2 mutations in the GDSEID motif was not entirely clear. Do the authors mean that because the yeast arrest at G2/M that these mutations are only relevant to CDC25 and not CKS1 or KAP function? The authors could remind readers what the different phenotypes are for the different genetic knockouts. Biochemical evidence supporting that those mutations specifically inhibit CDC25 and not the other proteins would be important to make that claim. Alternatively, perhaps the comparison is not so vital to the thrust of the paper, and the mechanistic explanation for the mutations could be moved instead to follow the TR-FRET data on CDC25 binding. In other words, the mutation data could be used simply to compliment the structural observation about where CDC25 binds, rather than to make a subtle (and not well substantiated) point about differences in CDC25 and CKS1 or KAP binding.*

We apologise for the confusion in this section. The CDK2 residues E208 and D206 equivalent to the two residues mutated in *S. pombe* DL2 and *cdc2E8* respectively both bind to CKS1 and CDC25A. We agree with the reviewer's point that though the residues are part of both CDK-protein interfaces we would need to show biochemical evidence to distinguish whether the observed DL2 and *cdc2E8* phenotypes result from CDK1's impaired ability to bind CDC25, or CKS1 or both proteins. To address the comment, we have followed the reviewer's suggestion and mentioned the mutation data to complement the structural observations of where CDC25 binds. We would suggest this text fits best in the concluding paragraph of the section entitled "The CDK2 GDSEID sequence mediates CDK2-protein interactions" after the new paragraph that describes the SPR experiments to evaluate the interaction of CDC25A with various CDK2 complexes. We have re-written and shortened this final paragraph (at the bottom of p19) to now read:

"In summary, it is evident from a comparison of CDK2 bound to KAP, CKS1 and CDC25A that the CDK2 GDSEID motif is an important mutual binding region for these proteins which confers selectivity by engaging different residues and hydrophobicity surface profiles. That it is a hotspot for protein binding provides the mechanistic explanation for the identification of yeast cell cycle mutants within the sequence^{71,72}. Further affinity and selectivity for CDC25A is provided by the N-terminal CDK2 binding regions, notably the G-loop and PSTAIRE helix. These regions appear less important for the binding of KAP and are irrelevant for CKS1 binding."

6) *There is some evidence in the literature that Cdk4 is phosphorylated in Y19 and that this phosphate is removed for G1/S progression, e.g. PMID: 7630405. There is less evidence that CDC25 is the phosphatase that removes the inhibitory phosphate in cells. The authors could more thoroughly discuss this literature, and if possible, test the prediction from the structure that CDK4/6 indeed binds with less affinity and/or dephosphorylate with less efficacy than CDK2.*

We thank the reviewer for mentioning this point. We do include PMID:7630405 (Ref 73, Terada et al.) and note that it is the only paper in the literature that provides evidence in support of CDK4 being phosphorylated on Tyr17.

To address the possibility that CDC25A dephosphorylates CDK4, we have used AlphaFold Multimer to assess potential complex formation between CDK4/6-cyclin D complexes and members of the CDC25 family. A summary of this analysis is now included as Supplementary Fig. 8, a section has been added to the Methods and the results are discussed in the text on p22/23 in a section now entitled “AlphaFold Multimer models to probe CDC25A substrate preferences”

.....However, CDK4-cyclin D complexes appear to be unlikely substrates for CDC25A based on sequence conservation. To further test this hypothesis, we employed AlphaFold Multimer⁷⁴ to assess the potential for CDK4-cyclin D3 to bind to CDC25A (Supplementary Fig. 8).

To benchmark our AlphaFold Multimer analysis, we first determined its ability to predict the structure of CDK2-cyclin A bound to CDC25A (Supplementary Fig. 8a). AlphaFold Multimer predicted this structure with significant accuracy: the five highest ranked models all indicated that CDC25A binds predominantly to CDK2, but they varied in the structural prediction for the CDC25A C-terminal tail. Notably, model 2 aligned to the cryo-EM determined structure with an overall RMSD of 0.8 Å.

We next modelled the structures of CDK4-cyclin D3-CDC25A (Supplementary Fig. 8b). The five highest ranked models consistently modelled CDC25A as binding to the cyclin subunit with no predicted interaction with CDK4. As illustrated by the Predicted Aligned Error (PAE) plots, confidence in each model was low. Thus, the AlphaFold Multimer models agree with the comparative sequence analysis and suggest that CDK4 is not a CDC25A substrate.

We hypothesize that where phosphorylation of CDK4 Tyr17 occurs, it may be dephosphorylated by other means. The detection of interactions between CDK4-cyclin D and CDC25 may result from CDK4 recognizing CDC25 proteins as substrates mediated through recruitment motifs within the CDC25 N-terminal sequence³⁹. The determination of the residues that mediate the interaction between the CDC25A catalytic domain and CDK2-cyclin A now provides a structural basis for further experiments to establish the importance of CDK4 inhibitory phosphorylation to regulation of its activity and the requirement for CDC25 phosphatases to modulate this post-translation modification.

7) Page 18, first full paragraph, first sentence beginning “Whilst”: Do the authors mean the last word of that sentence to be “CKS1” and not “KAP”?

We apologize for the confusion. We were trying to say that the binding of CKS1 is more comparable to the binding of KAP than to CDC25A. To clarify this point, we have split the sentence into two sentences:

“Whilst the recognition sites for CDC25A and CKS1 (PDB 1BUH) partially overlap at the GDSEID region, the exact nature of the interactions is different. The binding of CKS1 to CDK2 is more comparable to that of KAP binding to CDK2.”

8) It would be helpful if the authors chose to show 2D classes with comparable orientations

across the different conditions in Fig. 1a-d. For example, placing the similar orientations by column would allow the reader to gain a better appreciation of the differences in quality. Additional orientations could be shown with CHAPS to convey that more orientations were imaged in the presence of detergent. Also, do the authors intend to show the same 2D class panel in Fig. 1 and Sup. Fig 2?

Whilst we appreciate the reviewers comment here, it would be very difficult to align the 2D classes by similar orientation across the different conditions because most views observed in the presence of CHAPS were not present in the absence of CHAPS (hence our use of detergent to enhance the number of unique 2D views), therefore the classes across the different conditions are not comparable in view. We also think the chosen 2D classes in Figure 1 provide a strong representation of the 2D classes we observed, however, additional 2D classes for the CHAPS data can be found in Supplementary Fig. 3, which we believe addresses the comment for additional orientation. We have also modified the legend for Figure 1 to further convey the impact of CHAPS on particle distribution; the modified figure legend now reads:

“(a-c) Representative 2D class averages of particles of the CDK2-cyclin A-CDC25A complex picked from preliminary data using (a) blob picking, showing significant preferential orientation in which most 2D classes represented a front-on view of CDK2-CDC25A and (b) Topaz picking, which improved the number of unique 2D views, but preferential orientation persisted. (c) During high-resolution data collection, the use of CHAPS significantly enhanced the particle orientation distribution, as indicated by the wider variety of 2D class averages observed from (c) blob picked and (d) Topaz picked particles”

9) If possible, keeping the same orientation of the complex in Fig. 1 and 2 would be helpful. The two views of the complex in Figures 1 and 2 are very similar, but we have addressed this point by re-drawing Figure 2a so that the complex is in the same orientation as Figure 1e.

REVIEWER COMMENTS

Reviewer #1 (Remarks to the Author):

The authors addressed my comments very well.

Reviewer #2 (Remarks to the Author):

The authors have revised their manuscript according to the suggestions of three reviewers, which further strengthened it. The manuscript is now almost ready for publication; two technical comments remain:

1) The local resolution map shown in Fig. 1f looks odd, with very patchy variations in resolution and large resolution differences in immediately adjacent parts of the map (e.g. 2.2 Å right next to 2.6 Å in the C-lobe of CDK2 in the right-hand panel). This is unusual and suggests that the sub-volumes used for local FSC calculation by the algorithm were too small, making the resolution estimation noisy. While this issue does not affect the conclusions of the manuscript, it should still be corrected. This behaviour can be tuned in the options of the cryoSPARC local resolution job (Kernel width or adaptive window size should be increased). The authors should expect slightly worse local resolution that is more smoothly distributed, with the main variation not being patches, but (most likely) a gradient from the surface to the core of the particle.

2) The resolution at which the model vs. map FSC passes through the FSC = 0.5 threshold should be labelled in Supplementary Fig. 4e. It is currently difficult to discern the exact number (which is 2.8 Å, indicated in Supplementary Table 1).

Reviewer #3 (Remarks to the Author):

The authors have carefully considered reviewer comments, addressed questions well with more experiments and/or discussion where appropriate, and clarified several points to improve the manuscript. The study is of high quality and importance, and I recommend publication in Nature Communications.

Reviewer #4 (Remarks to Authors):

In this work, Rowland and authors present a cryo-EM structure of the complex between CDK2, cyclin A, and CDC25A. This complex structure shows a previously undetected C-terminal helix of CDC25A which sits in the interface between all 3 subunits. Further, HDX-MS analysis shows increased protection of both CDK2 and CDC25A residues upon binding, supporting the solved structure. The authors also use sequence and structural analysis to propose other CDK-cyclin binding partners to CDC25A. In my assessment, the work presented an interesting scientific contribution compatible with Nature Communications, though I provide the following comments and suggestions. I focus mainly on the HDX-MS as well as the sequence and structural analyses while deferring comments on solving the cryo-EM structure to the expertise of the other reviewers.

Major comments

1. The peptide difference plot for CDK2 shows 3 peptide regions that are significantly protected in the trimeric state. However, only one of these regions (Peptide 2, residues 196-220 with the GDSEID motif) is discussed. It is a bit hard to tell exactly with the current figure, but it appears that the other two significantly protected regions correspond to residues (though possibly they are adjacent residues) where the CDC25A C-terminal helix interacts with CDK2: the N-terminal α C helix (res 45-51) and the activation segment (res 145-172, possibly including phosphoT160). I would be interested to see a more in-depth discussion on the protected and deprotected peptides observed in the HDX-MS experiment.
2. The hypothesis that the C-terminal CDC25A helix may be highly dynamic when unbound seems intuitive and reasonable. It's disappointing that its protection/stabilization was not validated by the HDX-MS. It does appear to be slightly protected in the bound state, though not significantly. Data was only shown for 2 minutes of deuteration, though the methods state that experiments were performed at 0.5, 2, 10, or 30 minutes. Examining at shorter (than 2 minute) deuterium exposure time and/or including a time course experiment to further elucidate the dynamics and kinetics would help strengthen the highly dynamic helix hypothesis. I have left a reference at the end of this review that may be useful. It's a chapter of a book by David Weis that discusses in-depth the differences between highly dynamic and intrinsically disordered regions as seen by HDX.
3. The HDX and IM-MS methods and parameters follow good practices and typical use. The use of pepsin digestion, HDMSE, and CID should provide localized deuterium uptake with limited scrambling, though it would be nice to see more of this data and analysis. I would further encourage the authors to deposit the raw files in a proteomic database such as ProteomeXchange to promote data transparency.

Minor comments

1. Is there a reason the peptide coverage and difference plots in SI Fig 5 stop around residue 515 when the helix goes to residue 523 – were peptides just not identified?
2. In SI Fig 5 I think it would be helpful to state and label the “monomeric” CDK2 more explicitly as the CDK2-cyclin A complex which is referenced in the text to have been used in the experiments. Further, the method section states that HDX-MS was performed on 5 proteins/complexes – the 3 monomeric proteins, the CDK2-cyclin A complex, and the CDK2-cyclin A-CDC25A complex. However, data is only shown in SI Fig 5 for 4 proteins/complexes. Clarity on this would be appreciated.
3. The methods claim K_ds were calculated for the HTRF experiment, but they are not presented. It would be straightforward to include them in SI Fig 6 or its legend.
4. The sequence alignment section could be clarified by a brief in-text comment stating the residues discussed are involved in CDC25 recruitment, not in CDK-cyclin pairing. This may help explain why certain complexes like CDK1-cyclin B-CDC25A are suggested to bind while CDK2-cyclin B-CDC25A is not.
5. Minor technical typos: Top of page 4 “PDB 30P3” should be “PDB 3OP3”. Legend of SI Fig 5 should reference peptide coverage plots in panels a and d, not a and c. Legend of SI Fig 5 should reference CDK2-cyclin A, not just CDK2, as the “monomeric” state to match the text and figure titles. Table 1 labels b and c should be shifted down a row – while intuitive, it would be nice to reference these labels in the legend. In Table 1 it appears that CDK1 and CDK3 have 9, not 8, conserved residues.

Overall, Rowland and authors present an exciting structure for the CDK2-cyclin A-CDC25A complex. The structure not only identifies a novel helix in a key mitotic enzyme but provides information on potential allosteric drug target sites for cancer therapeutic development. I think the collected HDX-MS data could be further analyzed and presented. Additional HDX-MS experiments could strengthen the hypothesis that the C-terminal CDC25A helix is either disordered or highly dynamic, though it could be argued the additional experiments are beyond the scope of this study. This work lines up obvious several next steps which could additionally include a binding study to validate the proposed binding partners for the trimer complex formation. In general, I believe the work has significant merit, though the suggested adjustments could enhance its impact.

Reference

Weis, D. D. 2016. “Chapter 17: Analysis of Disordered Proteins by Hydrogen Exchange Mass Spectrometry”. Hydrogen exchange mass spectrometry of proteins: Fundamentals, methods, and applications. John Wiley & Sons, Ltd.

We would like to thank all three reviewers for taking the time to re-review the manuscript and Reviewer 4 for their review and further suggestions and comments. We have edited the submitted revised manuscript highlighting in blue the text changes made to address the reviewers' comments.

REVIEWER COMMENTS

Reviewer #1 (Remarks to the Author):

The authors addressed my comments very well.

Reviewer #2 (Remarks to the Author):

The authors have revised their manuscript according to the suggestions of three reviewers, which further strengthened it. The manuscript is now almost ready for publication; two technical comments remain:

1) The local resolution map shown in Fig. 1f looks odd, with very patchy variations in resolution and large resolution differences in immediately adjacent parts of the map (e.g. 2.2 Å right next to 2.6 Å in the C-lobe of CDK2 in the right-hand panel). This is unusual and suggests that the sub-volumes used for local FSC calculation by the algorithm were too small, making the resolution estimation noisy. While this issue does not affect the conclusions of the manuscript, it should still be corrected. This behaviour can be tuned in the options of the cryoSPARC local resolution job (Kernel width or adaptive window size should be increased). The authors should expect slightly worse local resolution that is more smoothly distributed, with the main variation not being patches, but (most likely) a gradient from the surface to the core of the particle.

We thank the reviewer for this suggestion. We appreciate that setting the width of the sub-volume used for local FSC calculation is a balance between accuracy in resolution and precision in locality, and therefore we usually allow the algorithm to set this automatically. However, based on the reviewer's comment, we have repeated the local resolution estimation with a larger kernel width and adaptive window. We believe this has smoothed out the resolution distribution, with a new local resolution range of 2.4 – 3.0 Å. We have updated Figure 1F on page 7, the figure legend now reads (changes highlighted in blue):

“(f) The local resolution within the complex varies from 2.4 to 3.0 Å (FSC 0.143) (map shown in same orientations as panel (e)).”

2) The resolution at which the model vs. map FSC passes through the FSC = 0.5 threshold should be labelled in Supplementary Fig. 4e. It is currently difficult to discern the exact number (which is 2.8 Å, indicated in Supplementary Table 1).

We have included the resolution at which the model-map FSC passes through the 0.5 threshold in the SI Figure 4e legend, for both masked (2.8 Å) and unmasked (3.0 Å) FSC curves. The figure legend on page 11 now reads:

“(e) A model-map FSC was generated using the comprehensive validation job in Phenix, showing the Fourier shell coefficient curve based on the model map with and without masking (orange and blue lines respectively). The intersections of the curves with FSC=0.5 are 2.8 Å with masking and 3.0 Å without masking.”

Reviewer #3 (Remarks to the Author):

The authors have carefully considered reviewer comments, addressed questions well with more experiments and/or discussion where appropriate, and clarified several points to improve the manuscript. The study is of high quality and importance, and I recommend publication in Nature Communications.

Reviewer #4 (Remarks to Authors):

In this work, Rowland and authors present a cryo-EM structure of the complex between CDK2, cyclin A, and CDC25A. This complex structure shows a previously undetected C-terminal helix of CDC25A which sits in the interface between all 3 subunits. Further, HDX-MS analysis shows increased protection of both CDK2 and CDC25A residues upon binding, supporting the solved structure. The authors also use sequence and structural analysis to propose other CDK-cyclin binding partners to CDC25A. In my assessment, the work presented an interesting scientific contribution compatible with Nature Communications, though I provide the following comments and suggestions. I focus mainly on the HDX-MS as well as the sequence and structural analyses while deferring comments on solving the cryo-EM structure to the expertise of the other reviewers.

Major comments

1. The peptide difference plot for CDK2 shows 3 peptide regions that are significantly protected in the trimeric state. However, only one of these regions (Peptide 2, residues 196-220 with the GDSEID motif) is discussed. It is a bit hard to tell exactly with the current figure, but it appears that the other two significantly protected regions correspond to residues (though possibly they are adjacent residues) where the CDC25A C-terminal helix interacts with CDK2: the N-terminal α C helix (res 45-51) and the activation segment (res 145-172, possibly including phosT160). I would be interested to see a more in-depth discussion on the protected and deprotected peptides observed in the HDX-MS experiment.

We thank the reviewer for highlighting these peptides. The protection observed in the CDK2 N-terminal peptide, residues 38-51 corresponds to the PSTAIRE helix and preceding loop. These residues bind the CDC25A C-terminal helix, and this interaction is discussed later in the text (pages 13-14). We have updated the text on page 10 to discuss this protection in the HDX analysis section. The text now reads;

“Within the CDK2 N-terminal lobe, a peptide spanning residues 38-51 (DTETEGVPSTAIRE) shows significant protection in the ternary complex. This peptide includes the PSTAIRE (\$\alpha\$ C) helix and preceding loop, which bind the CDC25A C-terminal helix (\$\alpha\$ 6) (described in detail below).”

The feature noted as slightly, but statistically significantly, protected in the activation segment corresponds to peptide 163-VVTLW-167. This peptide is not directly involved in

contacts with CDC25, though residues immediately N-terminal are, which may explain the observed protection. We now discuss this in the text on page 10:

“Slight, but statistically significant protection is also observed for a peptide within the CDK2 activation segment (residues 163-167, VVTLW). This peptide is not directly involved in CDC25A contact, but the immediately preceding residues (and activation segment as a whole) bind the CDC25A C-terminal helix (discussed below).”

2. The hypothesis that the C-terminal CDC25A helix may be highly dynamic when unbound seems intuitive and reasonable. It's disappointing that its protection/stabilization was not validated by the HDX-MS. It does appear to be slightly protected in the bound state, though not significantly. Data was only shown for 2 minutes of deuteration, though the methods state that experiments were performed at 0.5, 2, 10, or 30 minutes. Examining at shorter (than 2 minute) deuterium exposure time and/or including a time course experiment to further elucidate the dynamics and kinetics would help strengthen the highly dynamic helix hypothesis. I have left a reference at the end of this review that may be useful. It's a chapter of a book by David Weis that discusses in-depth the differences between highly dynamic and intrinsically disordered regions as seen by HDX.

We thank the referee for drawing our attention to the chapter by Weis, and we have investigated the order of the C-terminal helix further. As can be seen from the extent of exchange of C-terminal peptides evaluated across all of the timepoints collected, the levels of exchange are high in both complexed and unbound state, and do not appear to be time dependent. We conclude that the existing data lacks the temporal resolution to observe any difference in exchange between the bound and unbound states. Evaluating the refined atomic displacement parameters of atoms within the helix, we observe that temperature factors across the helix are high relative to the rest of the structure, with the exception of residues 512-517 that are directly involved in the contacts to CDK2 and cyclin A. We have amended the text to reference the high levels of exchange in both states, and to comment on the relative order of residues along the helix. The text at the bottom of page 10 now reads:

“Notably, the HDX-MS data did not reveal any significant protection for the C-terminal domain of CDC25A, which is surprising given the dominant feature of the CDC25A C-terminal ($\alpha 6$) helix in the cryo-EM map. Given the relatively distant nature of the interaction with the CDK2-cyclin A interface (the CDC25A C-terminal helix backbone sits ~ 11 Å away from the backbone structure of CDK2), it is hypothesized that the C-terminal helix is not substantially solvent-protected in the bound state. Additionally, high levels of exchange of C-terminal peptides were observed across all timepoints in both the unbound and bound states, which may be attributed to the structurally dynamic behaviour of this C-terminal region. This flexibility is supported by evaluation of the refined atomic displacement parameters of atoms within the helix, which reveals the temperature factors across the helix are high relative to the rest of the structure (with the exception of residues 512-517 that are directly involved in contacts to CDK2 and cyclin A). Therefore, it may not be possible to detect binding on the timescales measured by HDX-MS.”

3. The HDX and IM-MS methods and parameters follow good practices and typical use. The use of pepsin digestion, HDMSE, and CID should provide localized deuterium uptake with

limited scrambling, though it would be nice to see more of this data and analysis. I would further encourage the authors to deposit the raw files in a proteomic database such as ProteomeXchange to promote data transparency.

We thank the reviewer for this comment. We appreciate that there is significant potential for further HDX-MS analysis to provide insights into the dynamics and kinetics of CDC25A behaviour. However, we would request that given the focus of the manuscript these experiments are beyond the scope of the current manuscript. We can confirm that we have deposited the data in ProteomeXchange. We have added the following text in the Data availability statement on page 32 and included the reference.

“The mass spectrometry proteomics data have been deposited to the ProteomeXchange Consortium via the PRIDE ^{[86]} partner repository with the dataset identifier PXD050866”.

[86] Perez-Riverol Y, Bai J, Bandla C, Hewapathirana S, García-Seisdedos D, Kamatchinathan S, Kundu D, Prakash A, Frericks-Zipper A, Eisenacher M, Walzer M, Wang S, Brazma A, Vizcaíno JA (2022). The PRIDE database resources in 2022: A Hub for mass spectrometry-based proteomics evidences. *Nucleic Acids Res* 50(D1):D543-D552 (PubMed ID: 34723319).

The full submission and account details should the reviewer like to review the deposition are as follows:

Submission details: Project Name: Hydrogen-deuterium exchange mass spectrometry analysis of the CDK2-cyclin A-CDC25A complex
Project accession: PXD050866
Project DOI: Not applicable

Reviewer account details:
Username: reviewer_pxd050866@ebi.ac.uk
Password: R7MRWJZ6

We have also acknowledged the assistance of Dr A Calabrese in making the deposition.

Minor comments

1. Is there a reason the peptide coverage and difference plots in SI Fig 5 stop around residue 515 when the helix goes to residue 523 – were peptides just not identified?

We thank the reviewer for spotting that we didn't note this point in the MS. Peptides beyond 512 were not identified. The last identified peptide spans residues 490-512: (HHEDFKEDLKKFRTKSRTWAGEK). We have added the following text to the Supplementary Figure 5 legend:

“Peptides originating from CDC25A C-terminal to residue 512 were not identified.”

2. In SI Fig 5 I think it would be helpful to state and label the “monomeric” CDK2 more explicitly as the CDK2-cyclin A complex which is referenced in the text to have been used in the experiments. Further, the method section states that HDX-MS was performed on 5 proteins/complexes – the 3 monomeric proteins, the CDK2-cyclin A complex, and the CDK2-

cyclin A-CDC25A complex. However, data is only shown in SI Fig 5 for 4 proteins/complexes. Clarity on this would be appreciated.

We apologize for the confusion with the presentation of this data. Whilst we indicate analysis of monomeric CDK2 in the methods, we did not report this data as it was confounded by the effects of cyclin A binding, making it difficult to discern peptides affected only by CDC25A binding. Therefore, we chose to only analyze CDK2 in the dimeric CDK2-cyclin A complex vs CDK2-cyclin A-CDC25 complex; as indicated by the labelling of the peptide coverage plot in SI Figure 5d (Common peptides between States CDK2-cyclin A and CDK2-cyclin A-CDC25A). We have updated the methods (on page 30) to clarify which proteins/complexes were analyzed. The updated text now reads:

“Proteins (pY15pT160CDK2, cyclin A CDC25A(Cys431Ser), pY15pT160CDK2-cyclin A and pY15pT160CDK2-cyclin A-CDC25A(Cys431Ser)) were diluted to 10 μM in equilibration buffer (40 mM HEPES pH 7.4, 150 mM NaCl, 1mM TCEP.HCl) prior to analysis.”

3. The methods claim Kds were calculated for the HTRF experiment, but they are not presented. It would be straightforward to include them in SI Fig 6 or its legend.

We thank the reviewer for this suggestion. We have added the Kd value ($0.11 \pm 0.02 \mu\text{M}$) to the figure legend).

4. The sequence alignment section could be clarified by a brief in-text comment stating the residues discussed are involved in CDC25 recruitment, not in CDK-cyclin pairing. This may help explain why certain complexes like CDK1-cyclin B-CDC25A are suggested to bind while CDK2-cyclin B-CDC25A is not.

We have modified the top full paragraph on page 20 to clarify that sequence analysis was to assess the conservation of residues at key CDC25A binding regions. This paragraph now reads:

“This cryo-EM complex provides a template from which to assess the sequence differences between members of the CDK, cyclin and CDC25 families at key residues that mediate the interaction between CDC25A and CDK2-cyclin A to identify potential partner preferences (Table 1, Supplementary Fig. 7).”

We have also clarified this point in the last paragraph on page 22 to rationalise that some CDK-cyclin complexes are proposed to bind to CDC25A whilst others are not. This paragraph now reads;

“This sequence conservation analysis of critical residues involved in CDC25A recruitment suggests that CDK1/2-cyclin A, CDK1-cyclin B and CDK2/3-cyclin E complexes are suitable binding partners for CDC25A”

The Table 1 legend also clarifies this point which now reads:

Selected residues of (a) CDK2 (b) cyclin A and (c) CDC25A identified from the cryo-EM structure that mediate complex formation. The conservation of these residues across

relevant CDKs, cyclins and CDC25 isoforms are indicated in purple. Complete sequence alignments are provided in Supplementary Fig. 7. “

5. Minor technical typos: Top of page 4 “PDB 30P3” should be “PDB 3OP3”. Legend of SI Fig 5 should reference peptide coverage plots in panels a and d, not a and c. Legend of SI Fig 5 should reference CDK2-cyclin A, not just CDK2, as the “monomeric” state to match the text and figure titles. Table 1 labels b and c should be shifted down a row – while intuitive, it would be nice to reference these labels in the legend. In Tabel 1 it appears that CDK1 and CDK3 have 9, not 8, conserved residues.

We apologize for these errors and thank the reviewer for bringing them to our attention. We have amended the text appropriately;

- (i) PDB 30P3 has been corrected to POP3 at the top of page 4.
- (ii) The referencing of peptide coverage plots in SI figure 5 has been corrected from a and c, to a and d.
- (iii) The legend of SI figure 5 has been updated to reference CDK2-cyclin A rather than monomeric CDK2 and now reads;

“Peptide coverage plot for (a) CDC25A and (d) CDK2 in the CDK2-cyclin A complex. (b,e) Peptide difference plots for the 2 min deuterium incubation time point for (b) CDC25A and (e) CDK2”
- (iv) The labels on Table 1 have been adjusted.
- (v) The legend for Table 1 has been changed to reference the labelling. The legend now reads;

“Selected residues of (a) CDK2 (b) cyclin A and (c) CDC25A identified from the cryo-EM structure that mediate complex formation. The conservation of these residues across relevant CDKs, cyclins and CDC25 isoforms are indicated in purple. Complete sequence alignments are provided in Supplementary Fig. 7”
- (vi) The number of conserved residues for CDK1 and CDK3 in Table 1 has been corrected from 8 to 9.

Overall, Rowland and authors present an exciting structure for the CDK2-cyclin A-CDC25A complex. The structure not only identifies a novel helix in a key mitotic enzyme but provides information on potential allosteric drug target sites for cancer therapeutic development. I think the collected HDX-MS data could be further analyzed and presented. Additional HDX-MS experiments could strengthen the hypothesis that the C-terminal CDC25A helix is either disordered or highly dynamic, though it could be argued the additional experiments are beyond the scope of this study. This work lines up obvious several next steps which could additionally include a binding study to validate the proposed binding partners for the trimer complex formation. In general, I believe the work has significant merit, though the suggested adjustments could enhance its impact.

We thank the reviewer for their supportive summary of our experiments. We have sought to incorporate changes that enhance the analysis with the available data, drawing, where appropriate, on orthogonal measures of protein dynamics such as analysis of the protein temperature factors. We appreciate that there is much more analysis supported by further experiments that could be carried out to analyze CDC25 behavior but would respectfully suggest that this work is beyond the scope of the current manuscript.

Reference

Weis, D. D. 2016. "Chapter 17: Analysis of Disordered Proteins by Hydrogen Exchange Mass Spectrometry". Hydrogen exchange mass spectrometry of proteins: Fundamentals, methods, and applications. John Wiley & Sons, Ltd.

REVIEWERS' COMMENTS

Reviewer #2 (Remarks to the Author):

The authors addressed the two minor issues raised in the previous round of review. I have no further comments.

Reviewer #4 (Remarks to the Author):

The authors have thoroughly addressed the previous comments and revised the text in several places. I particularly appreciate their willingness to provide further explanations and discussions on the HDX and sequence analysis sections, as well as their promotion of data transparency through depositing their mass spectrometry results in an appropriate database. They present a strong manuscript that I would recommend for publishing in Nature Communications.

I have just only minor adjustment to suggest- In Supplementary Figure 5e, the listed counts of deprotected, non-significant, and protected peptides are inaccurate. It appears that they were accidentally copied from panel b in the figure and should be updated to reflect the appropriate data

We would like to thank all reviewers for taking the time to re-review the manuscript and Reviewer 4 for their suggestion. We have edited the submitted revised manuscript to address this comment.

REVIEWERS' COMMENTS

Reviewer #2 (Remarks to the Author):

The authors addressed the two minor issues raised in the previous round of review. I have no further comments.

Reviewer #4 (Remarks to the Author):

The authors have thoroughly addressed the previous comments and revised the text in several places. I particularly appreciate their willingness to provide further explanations and discussions on the HDX and sequence analysis sections, as well as their promotion of data transparency through depositing their mass spectrometry results in an appropriate database. They present a strong manuscript that I would recommend for publishing in Nature Communications.

I have just only minor adjustment to suggest- In Supplementary Figure 5e, the listed counts of deprotected, non-significant, and protected peptides are inaccurate. It appears that they were accidentally copied from panel b in the figure and should be updated to reflect the appropriate data.

We thank the reviewer for spotting this and apologize for this mistake. We have updated Supplementary Figure 5e to display the correct number of deprotected, non-sig and protected residues.